# Targeting immune checkpoints potentiates immunoediting and changes the dynamics of tumor evolution

Mirjana Efremova[1], Dietmar Rieder[1], Victoria Klepsch[2], Pornpimol Charoentong[1], Francesca Finotello [1], Hubert Hackl [1], Natascha Hermann-Kleiter[2], Martin Löwer[3], Gottfried Baier [2], Anne Krogsdam[1] & Zlatko Trajanoski [1]

The cancer immunoediting hypothesis postulates a dual role of the immune system: protecting the host by eliminating tumor cells, and shaping the tumor by editing its genome. Here, we elucidate the impact of evolutionary and immune-related forces on editing the tumor in a mouse model for hypermutated and microsatellite-instable colorectal cancer. Analyses of wild-type and immunodeficient RAG1 knockout mice transplanted with MC38 cells reveal that upregulation of checkpoint molecules and infiltration by Tregs are the major tumor escape mechanisms. Our results show that the effects of immunoediting are weak and that neutral accumulation of mutations dominates. Targeting the PD-1/PD-L1 pathway using immune checkpoint blocker effectively potentiates immunoediting. The immunoediting effects are less pronounced in the CT26 cell line, a non-hypermutated/microsatellite-instable model. Our study demonstrates that neutral evolution is another force that contributes to sculpting the tumor and that checkpoint blockade effectively enforces T-cell-dependent immunoselective pressure.

[1] Biocenter, Division of Bioinformatics, Medical University of Innsbruck, Innsbruck, Austria. [2] Division of Translational Cell Genetics, Medical University of Innsbruck, Innsbruck, Austria. [3] TRON –Translational Oncology at the University Medical Center of the Johannes Gutenberg University gGmbH, Mainz, Germany. Correspondence and requests for materials should be addressed to A.K. (email: anne.krogsdam@i-med.ac.at) or to Z.T. (email: zlatko.trajanoski@i-med.ac.at)

The concept of cancer immunosurveillance, i.e., that lymphocytes can recognize and eliminate tumor cells, was proposed almost 50 years ago[1]. The definitive work supporting the existence of this process was published 30 years later by the Schreiber lab[2]. In this seminal work, an elegant experiment was carried out using a mouse model lacking the recombination activating gene 2 (RAG2), which encodes a protein involved in the initiation of V(D)J recombination during B- and T-cell development. RAG2-deficient mice, which are viable but fail to produce mature B or T lymphocytes[3], developed sarcomas more rapidly and with greater frequency than genetically matched wild-type controls[2]. Moreover, tumors derived from those mice were more immunogenic than those from wild-type mice[2]. These findings led to the development of the refined cancer immunosurveillance concept: the cancer immunoediting hypothesis[4]. The cancer immunoediting postulates a dual role of the immunity in the complex interactions between tumor and host; the immune system, by recognizing tumor-specific antigens, not only protects the host through elimination of tumor cells, but can also sculpt the developing tumor by editing the cancer genome, thereby producing variants with reduced immunogenicity.

Cancer immunoediting is more difficult to study in humans, but clinical data from patients with severe immunodeficiencies is supporting the notion that this process also exists in humans[5]. Indirect evidence for the existence of immunoediting in some cancers was provided by calculating the ratio of observed and predicted neoantigens, i.e., tumor antigens derived from mutated proteins[6]. Using a similar approach, we recently provided additional data supporting the existence of immunoediting in microsatellite-instable (MSI) colorectal cancer (CRC)[7]. However, as we recently showed in a pan-cancer genomic analysis, the composition of the intratumoral immune infiltrates is highly heterogeneous and changing during tumor progression[8] and hinders the distinction of genetic, immune, and other evasion mechanisms. Over and above these mechanistic questions on tumor progression, there is an urgent need to investigate cancer immunoediting also in the context of cancer immunotherapy. Cancer immunotherapy with checkpoint inhibitors like anti-CTLA-4 or anti-PD-1/-PD-L1 antibodies are showing remarkable clinical responses[9]. However, one of the biggest challenges is intrinsic resistance to immunotherapy and the development of resistant disease after therapy, i.e., acquired resistance to immunotherapy. As many patients with advanced cancers are now receiving immunotherapy, elucidating the role of cancer immunoediting as a potential mechanism of acquired resistance to immunotherapy[10] is of utmost importance.

Surprisingly, despite the recognition of the cancer immunoediting process and the widespread use of both mouse models and next-generation sequencing (NGS) technologies, the impact of immunoediting on the cancer genome has not been well characterized. Cancer immunoediting was investigated in a mouse model of sarcoma using NGS of the tumor exome and algorithms for predicting neoantigens[11]. This sarcoma model showed that immunoediting can produce tumor cells that lack tumor-specific rejection antigens, but how this finding translates into common human malignancies remained unclear. Later, two widely used tumor models, a CRC cell line MC38 and a prostate cancer cell line TRAMP-C1, were used to identify immunogenic tumor mutations by combining NGS and mass spectrometry[12]. However, as neither longitudinal samples of wild-type or immunodeficient mice nor checkpoint blockade was applied, two major questions remain unanswered: (1) To what extent is T-cell-dependent immunoselection sculpting the cancer genome? (2) How is immunotherapy with checkpoint blockers modulating immunoediting? Quantitative evaluation of immunoediting during tumor progression, as well as following therapeutic intervention using

checkpoint blockers could not only provide novel mechanistic insights, but might also inform immunotherapeutic strategies that could potentially be translated into the clinic.

We therefore designed a study to investigate immunoediting of an epithelial cancer genome using wild-type and immunodeficient mice, NGS, and analytical pipelines to process and analyze the data. We first characterize the genomic and transcriptomic landscape of the mouse colon adenocarcinoma cell line MC38 (mouse colon #38) that was induced by the subcutaneous injection of dimethylhydrazine in C57Bl/6 mice[13], and show that this cell line is a valid model for hypermutated/MSI CRC. We then carry out experiments with wild-type and immunodeficient RAG1$^{-/-}$ mice with transplanted tumors and analyze longitudinal samples with respect to the genomic landscape and the immunophenotypes of the tumors. The results show the extent of immunoediting of the cancer genome in this model in relation to other selection processes. Finally, we perform experiments with anti-PD-L1 antibodies using the MC38 cell line and another CRC cell line which is a model for non-hypermutated/MSI- CRC (CT26 (Colon Tumor #26)) and show how targeting the PD-1/PD-L1 pathway modulates immunoediting.

## Results

**Immunogenomic and transcriptomic characterization of MC38 cell line.** Functional studies on immunoediting require genetic tools and controls afforded by mouse studies. As immunoediting has not been quantified using mouse epithelial cancers so far, we designed experiments with transplanted tumors using the murine MC38 cell line. The MC38 murine CRC cell line is derived from a grade-III adenocarcinoma that was chemically induced in a female C57BL/6 mouse and used since then as a transplantable mouse tumor model[14]. Several studies have shown that the cell line is immunogenic and can be used as a model for investigating anticancer immunity and immunotherapy[15–18]. To characterize the genome and transcriptome of the MC38 cell line, we performed whole-exome sequencing, SNP array analysis, and RNA sequencing (Fig. 1a). We identified 5931 somatic mutations of which 2743 were nonsynonymous (2585 missense, 158 stop-gained) and 354 indels (Fig. 1b). Of the 5931 SNVs, the majority (4775) were transversions, of which most (2759) were C > A/G > T. Human hypermutated CRC tumors containing POLE mutations showed increased proportions of C > A/G > T and T > G/A > C transversions[19,20]. In contrast, it has been shown that the mouse CT26 cell line shows predominantly C > T/G > A SNVs[21], similar to primary human non-hypermutated CRC tumors[22]. Analysis of the MC38 data using previously published mutational signatures[19] revealed a mutational profile consisting of a combination of signatures, including the signature for DNA mismatch-repair (MMR) deficiency (Supplementary Figure 1).

We investigated whether known CRC driver mutations are also present in MC38. We found missense mutations in TP53, PTEN, and mutations in the tumor growth factor (TGF) beta pathway (SMAD2, SMAD4, ACVR2A, TGFB2, but not TGFBR2). BRAF was also mutated, which is frequently associated with the MSI-high phenotype[23]. KRAS was not mutated and there was only one intron mutation in APC. However, there was a truncating mutation in AXIN2 which is known to regulate β-catenin in the Wnt signaling pathway. The frequent mutations in SOX9 and ARID1A[22] were also present in the MC38 cell line. SOX9 is a transcription factor that inhibits Wnt signaling[24] and has a role in regulating cell differentiation in the intestinal stem cell niche[25], whereas ARID1A is involved in suppressing MYC transcription[26]. Four driver mutations of the MC38 cell line correspond to known hotspots in human CRC, albeit at different positions[27] (Supplementary Data 1).

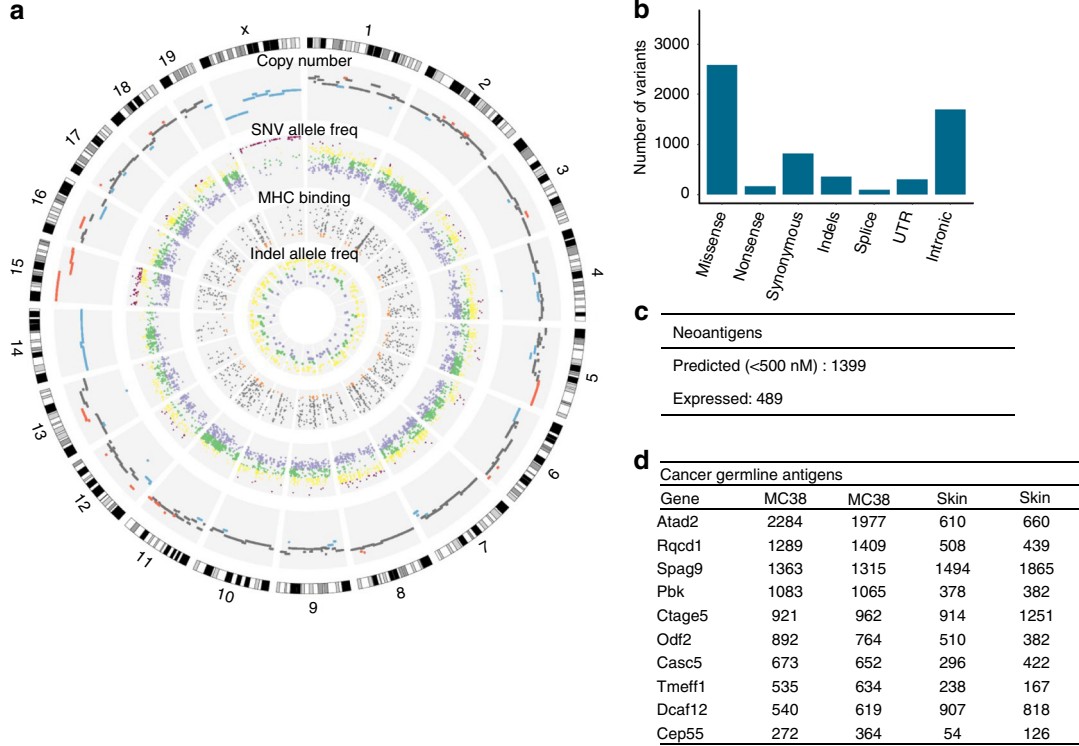

**Fig. 1** Immunogenomic and transcriptomic characterization of the mouse MC38 cell line. **a** Circos plot showing (outer to inner): cytogenetic bands in black, gray, and white. Track 1: DNA copy number log ratio values. Dark gray: diploid; Red: amplification (log ratio > 0.25); Blue: deletion (log ratio < −0.25). Track 2: Point mutations, plotted based on the variant allele frequency. Inner is frequency 0, outer is 100. Colors are purple (0–40), green (40–60), yellow (60–80), and pink (80–100). Track 3: Predicted MHC binding IC50 scores for the nonsynonymous mutations. Mutations with the highest binding affinity are colored orange (IC50 < 50). Track 4: insertions and deletions colored according to their allele frequency. **b** Number of mutations in MC38 classified by type. **c** Number of predicted and expressed neoantigens in MC38. **d** Known germline antigens with the highest expression in MC38. The expression values are in normalized counts

In a previous large-scale genomic study of human colorectal samples three subtypes of colorectal cancer were identified[22]: (1) microsatellite stable tumors (MSS), (2) tumors with MSI due to a DNA MMR system deficiency, and (3) hypermutated tumors that harbor mutations in the exonuclease (proofreading) domain of the DNA polymerases Pol δ (POLD1) and Pol ε (POLE). The MC38 data also showed mutations in the MMR gene MSH3, as well as in POLD1, indicating that the MC38 cell line is a valid model to study human MSI and hypermutated CRC. Both MSI and hypermutated CRC were reported to have better prognosis, higher infiltration of CD8+ T cells and respond well to checkpoint blockade therapy[28], likely due to the high number of neoantigens.

We then characterized copy number variants of the MC38 cell line using exome sequencing and SNP arrays. The analysis of the copy number profiles inferred from the exome-sequencing data using hidden Markov model algorithm (see Methods) and from the SNP array data were concordant and showed a mostly diploid genome, with some regions of amplifications and deletions (Fig. 1a). We identified amplifications in the regions that contain the MYC and ERBB2 genes. Finally, we carried out transcriptomic analysis of the MC38 cell line in comparison with normal skin tissue. The transcriptomic data were used to: (1) identify pathways that were up- or downregulated in the cell line, and (2) to identify expressed tumor antigens, including neoantigens (identified using exome-sequencing data and a prediction algorithm as previously described[29]) and cancer-germline antigens (CGAs). The latter are tumor antigens that are considered to be tumor-specific as these molecules are expressed only in germline cells and in tumor cells. Pathway enrichment analysis

identified pathways related to cell cycle, DNA replication, DNA repair, and metabolism of nucleotides (Supplementary Figure 2).

With respect to the tumor antigens, we identified a large number of expressed neoantigens (Fig. 1c) and expressed CGAs (Fig. 1d), which provide evidence for the immunogenicity of this model. Of the 2743 amino-acid changes (missense and stop codon) in MC38, 1399 neoantigens were predicted to strongly bind to the C57Bl/6 major histocompatibility molecules (MHC) class I molecules H2-Kb and H2-Db with < 500 nM, and of these, 489 were in expressed genes. In addition, several CGAs were highly expressed in MC38 including TAD2, RQCD1, SPAG9, PBK, CTAGE5, CASC5, and CEP55, which were also found to be expressed in the CT26 cell line[21]. It is noteworthy that these CGAs were also expressed in the skin samples.

Thus, the characterization of the genomic and transcriptomic landscape of the CRC MC38 cell line demonstrates its validity as a model for hypermutated and/or MSI colorectal cancer.

**Upregulating checkpoints is a tumor escape mechanism in MC38 cell line**. In our mouse model used to recapitulate the process of cancer immunoediting, MC38 cells were subcutaneously injected into wild-type C57Bl/6 and immunodeficient RAG1−/− mice. The tumor growth was monitored regularly and samples were collected at predefined time points and subjected to detailed analysis using FACS, exome and RNA sequencing, and SNP array analysis (Fig. 2a). As expected, the tumor growth was significantly faster (two-tailed unpaired Student's $t$-test, $p$-value of 0.019) in RAG1−/− mice compared with the wild-type mice (Fig. 2b).

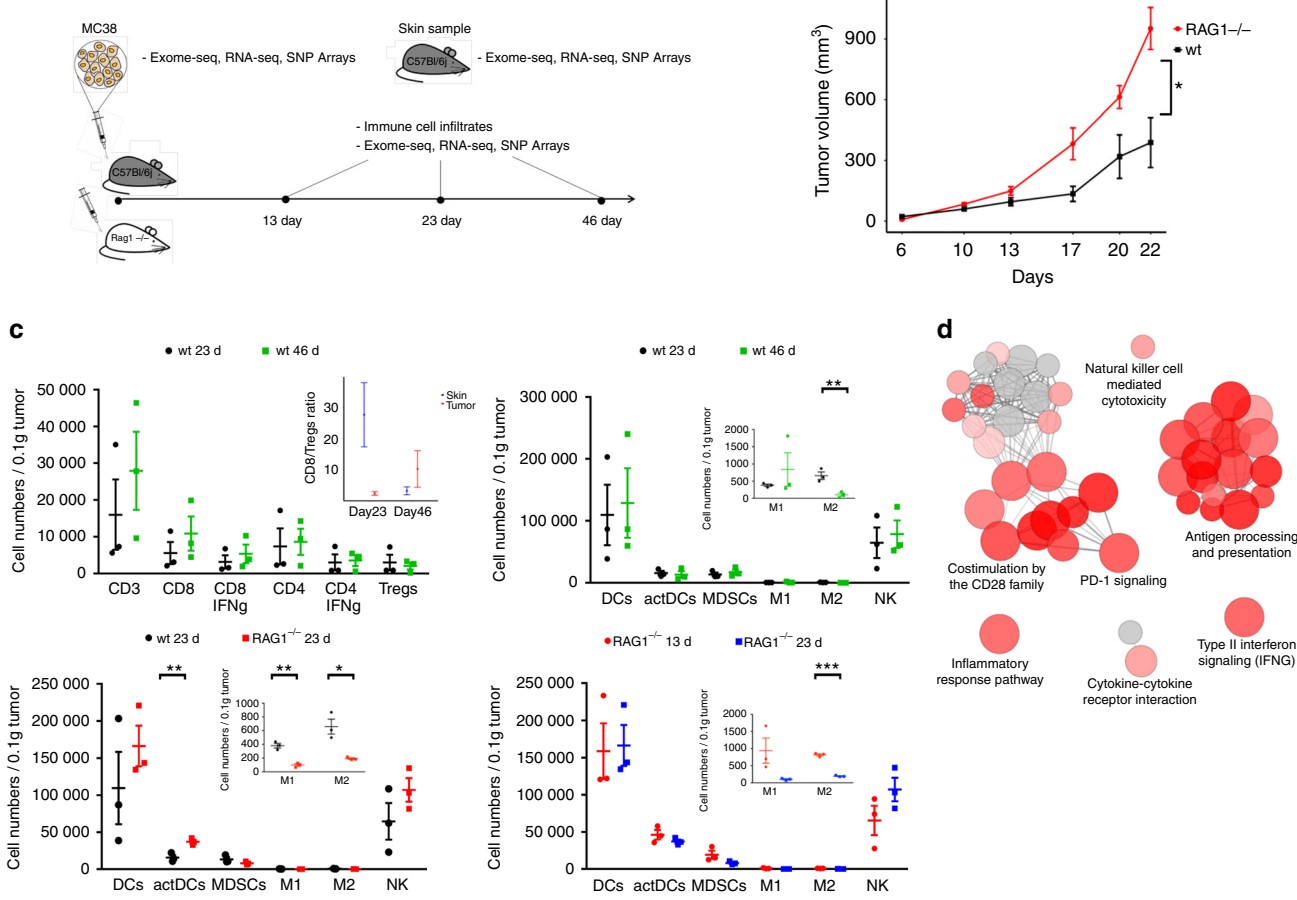

**Fig. 2** Tumor progression and tumor-infiltrating immune cells in wild-type and RAG1$^{-/-}$ mice. **a** Schematic diagram of the experimental setup. **b** Tumor growth curves of $5 \times 10^4$ MC38 cells inoculated into C57Bl/6 wild-type ($n = 10$) and RAG1$^{-/-}$ mice ($n = 4$). **c** Tumor-infiltrating lymphocytes in wild-type and RAG1$^{-/-}$ mice analyzed by flow cytometry. **b**, **c** The data are presented as the mean $\pm$ SEM, analyzed by a two-tailed unpaired Student's $t$-test. Statistical significance is indicated as *$p < 0.05$, ** $p < 0.01$, *** $p < 0.001$. **d** Enriched functions and pathways of the significantly differentially expressed genes in tumors of the wild-type vs RAG1$^{-/-}$ mice taken at day 23. The network is created using ClueGO. The pathways are functionally grouped based on the kappa score and the most significant term of each group is highlighted. The size of the nodes shows the enrichment significance after Bonferroni correction

FACS analysis revealed infiltration of both innate and adaptive immune cells including CD8$^+$ T cells, (natural killer) NK cells, and classically activated (M1) macrophages in wild-type mice, that increased with time, although not significantly (Fig. 2c). RNA expression profiles revealed higher expression of chemoattractant molecules such as CXCL9 and CCL5 in wild-type mice in comparison with immunodeficient mice (Supplementary Figure 3). However, despite the presence of tumor-infiltrating lymphocytes in the slow-growing tumors, the adaptive immune system failed to eliminate the tumors. Tumors may utilize several mechanisms of escape such as antigen loss, upregulation of inhibitory molecules, downregulation of MHC, or establishment of an immunosuppressive environment. The CD8/Treg cell ratio, which is a surrogate marker for a suppressive tumor micro-environment, was higher in the skin samples compared with the tumor samples at day 23 (Fig. 2c), suggesting that one escape mechanism in this model is the presence of immunosuppressive cells. The number of myeloid-derived suppressive cells and Tregs were comparable in both time points in wild-type mice, whereas the alternatively activated (M2) macrophages were significantly reduced (two-tailed unpaired Student's $t$-test, $p$-value of 0.009). The tumor progression in wild-type samples was associated with upregulation of immunoinhibitory genes, including PD-1, CTLA-4, TIM3, and LAG3 (Supplementary Figure 4). MC38 cells expressed low levels of PD-L1, whereas PD-L1 was slightly

upregulated in RAG1$^{-/-}$ and more in wild-type mice. Our finding is in accordance with previous studies showing that PD-L1 expression of MC38-transplanted tumors increases as a result of exposure to inflammatory cytokines such as interferon gamma (IFNγ), which is sufficient for tumor escape and immune evasion[30,31]. The upregulation of PD-L1 is likely due to the phenotypic plasticity and not positive selection of high PD-L1 clones. Pathway analysis of the differentially expressed genes in wild-type vs RAG1$^{-/-}$ tumors showed upregulation of several immune processes related to activation of an adaptive immune system response such as costimulation by CD28, PD-1 signaling, antigen processing and presentation, NK cell-mediated cytotoxicity, TCR signaling and IFNγ signaling (Fig. 2d and Supplementary Figure 5a) Downregulated pathways and GO terms included processes related to cell cycle, DNA replication, and TNF signaling (Supplementary Figure 5b).

These data indicate that two tumor escape mechanisms are activated in this model: infiltration of immunosuppressive Tregs and upregulation of inhibitory genes.

**Neutral evolution outweighs immunoselection.** Tumor progression is an evolutionary process under Darwinian selection[32], a characteristic that has been attributed as the primary reason of therapeutic failure, but also as a feature that holds the key to more

effective tumor control. At the time of detection, a tumor has acquired novel somatic mutations of which only a small subset (called driver mutations) provide an evolutionary advantage. The immune system also exerts an evolutionary pressure, through a T-cell-dependent immunoselection process by acting on tumor clones that display strong rejection antigens[11], and to some extent by T-cell-independent immunoselection through M1 macrophages, IFNγ, and NK cells[33]. In addition to the ongoing evolutionary and immune-related clonal selection, a recent study using a theoretical model demonstrated the occurrence of neutral evolution during tumor development[34]. According to this model, tumor heterogeneity in some cancers, including CRC, can be explained by neutral expansion and the accumulation of passenger mutations without selective sweeps.

To elucidate the impact of immunoselection on the progressing tumor, we used exome sequencing to identify nonsynonymous mutations and a MHC class I binding algorithm to predict candidate neoantigens. The number of shared mutations was similar between the biological replicates, suggesting that the sampling bias is rather small (Supplementary Figures 7 and 8). Further analysis of the exome-sequencing data showed a high number of mutations that were shared between the MC38 cell line and the two consecutive time points in both, wild-type (2299) and RAG1$^{-/-}$ (2372) samples (Fig. 3a).

According to the cancer immunoediting hypothesis, the immune system can sculpt the developing tumor by editing the cancer genome, thereby modifying the heterogeneity of the tumor: strong immunoediting would render tumors more homogeneous by eradicating immunogenic clones. In order to analyze the heterogeneity of the tumors during progression, exome-sequencing data and SNP array data was used to estimate cancer cell fractions (CCF) of all point mutations and, subsequently, tumor heterogeneity. Analyses of the tumor heterogeneity did not reveal large differences during progression in both, wild-type and RAG1$^{-/-}$ samples (Fig. 3b). Strikingly, the analyses showed that the variant allele frequencies (VAF) of the majority of the mutations did not change with time in both the wild-type and in the RAG1$^{-/-}$ mouse. On average, 3–5% of the mutations in the wild-type and in the RAG1$^{-/-}$ samples did not change their VAF (Supplementary Table 1) suggesting that neutral evolution, rather than Darwinian evolution, is driving the tumor growth in this model.

We then characterized the neoantigens using exome-sequencing data (to derive somatic mutations), RNA-sequencing data (for filtering expressed mutations) and an algorithm for predicting peptide-MHC binding affinity (see Methods). In order to identify immunogenic mutations, we selected the expressed neoantigens with the highest binding affinity (IC$_{50}$ < 500 nM). In a previous study with the MC38 cell line, seven mutant peptides were identified, using mass spectrometry, of which two elicited a T-cell response[12]. In our analysis, five out the seven peptides were predicted and four of them were detectable from the RNA expression data (Fig. 3c) The large impact of neutral evolution was evident also in the Venn diagrams of the number of neoantigens (Fig. 3d). The number of identified newly generated neoantigens was comparable in all samples (75 and 66 for the wild-type samples at day 23 and 46) and was higher than the potentially lost or targeted neoantigens (31 and 24 in the wild-type samples). As the key value to understand immunoediting is the ratio between expressed neoantigens and total number of mutations, we calculated these values, and show that the ratio was similar across all samples (Supplementary Figure 5a). Finally, to exclude the possibility that clones may have lost neoantigen-generating mutations, we determined the degree of loss of heterozygosity (LOH) in the samples. Although the number of events increased in the transplanted tumors in wild-type and in immunodeficient mice

(Supplementary Figure 5b), there were no LOH events at the genomic positions of the neoantigens, suggesting that no neoantigens were lost owing to LOH.

We then focused our analysis on the tumor samples taken at the same time point, day 23, for wild-type and RAG1$^{-/-}$ samples and considered neoantigens found both in the MC38 cell line and in at least one of the RAG1$^{-/-}$ tumors (Fig. 3e). There were 409 neoantigens shared by the wild-type and RAG1$^{-/-}$ tumors, and the MC38 cell line samples. About 6% of the neoantigens (23 out of 409) were detectable only in RAG1$^{-/-}$ tumors (Supplementary Data 2), out of which 21 were derived from mutations not detected or eliminated in the wild-type tumors. Only two neoantigens were lost because of low expression. The small number of lost neoantigens imply that the impact of the T-cell-dependent immunoediting in this model is rather modest. In addition, a similar number of neoantigens (14) was detectable only in wild-type tumors, suggesting that these neoantigens were edited by T-cell-independent mechanisms. Upregulation of genes related to NK cell-mediated toxicity and IFN signaling further supports this observation (Supplementary Figure 6a). Analysis of the downregulated transcripts revealed genes related to DNA replication and cell cycle (Supplementary Figure 6b).

Heterogeneity analysis showed that all MC38-derived tumors as well as the MC38 cell line, were similarly heterogeneous (Fig. 3f, g). To infer how the clonal composition changes between samples, we used a Bayesian Dirichlet process to cluster clonal and subclonal mutations. The results showed that the clonal and subclonal clusters were on the leading diagonal of the plots indicating there was no change in the mutational profile and the clonal/subclonal composition between any two samples (Fig. 3f). There was a large percentage of clonal mutations (60–70%) both in the MC38 and in the individual tumor samples (Fig. 3h and Supplementary Figures 7, 8). This was evident also from the VAF plots of the mutations found in diploid regions (Supplementary Figure 9).

Overall, the results suggest that the clonal dynamics of this cancer cell line over time is not dominated by strong Darwinian selection, but rather follows neutral evolution.

**Targeting the PD-1/PD-L1 pathway potentiates immunoediting.** By targeting the PD-1/PD-L1 axis to generate a strong immunological pressure, we next investigated the resulting impact on the cancer genome, on the neoantigen landscape, and on the tumor heterogeneity. It was previously shown that MC38 responds to various immunotherapies[16,18,35,36]. In order to identify neoantigens that would be potential targets of T cells activated by checkpoint blockade therapy, wild-type C57Bl/6 mice were treated with anti PD-L1 antibodies or IgG2b antibodies as control. Treatment was started 1 day after tumor inoculation and then repeated every 3–4 days. Samples from six tumors treated with anti-PD-L1 and six tumors treated with IgG2b were taken on day 14. Three samples of each group were used for exome sequencing, and three for RNA-sequencing.

Treatment with anti-PD-L1 antibodies reduced tumor growth in the treated mice compared with the controls by 65% (Fig. 4a), which is in line with previous studies showing that MC38 responds well to PD-1/PD-L1 blockade therapy[37,38]. This was further reflected in the RNA-sequencing data, which showed a strong upregulation of IFNγ, perforin, and granzyme A and B (GZMA and GZMB), as well as different immunomodulators and MHC molecules (Supplementary Figure 10). GO and pathway analysis showed upregulation of immune-related processes such as PD-1 signaling, chemokine signaling, cytokine-cytokine receptor interaction, and NK cell-mediated cytotoxicity (Fig. 4b and Supplementary Figure 11). Hence, blocking of the PD-1/PD-

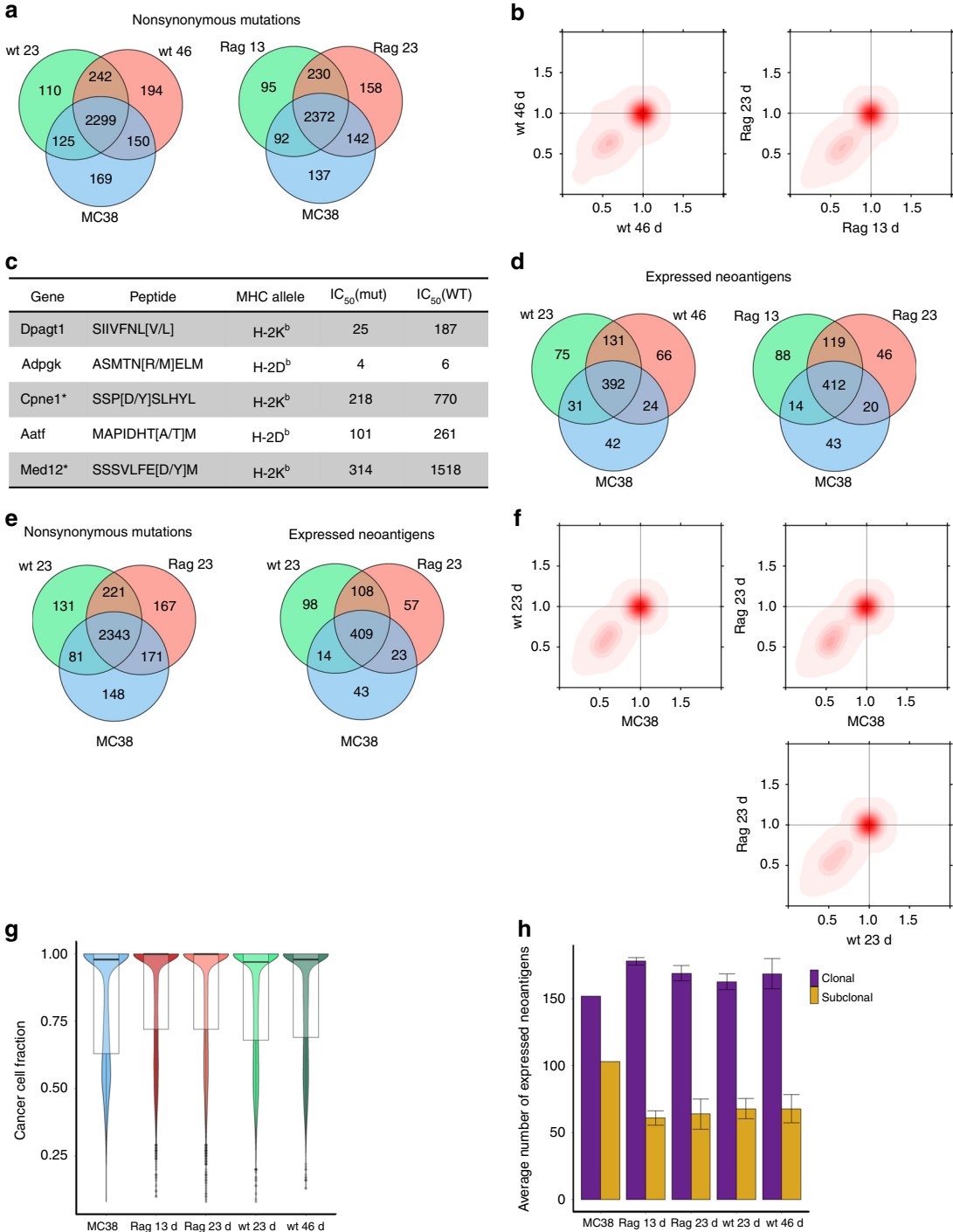

**Fig. 3** Genomic and immunogenomic analyses of progressing tumors in wild-type and RAG1-/- mice. **a** Shared nonsynonymous mutations between MC38, and wild-type and RAG1$^{-/-}$ samples during progression. Mutations found in at least one sample from the same type are considered. **b** Two-dimensional density plots showing the clustering of the cancer cell fractions of all mutations shared between two samples; increasing intensity of red indicates the location of a high posterior probability of a cluster. **c** Immunogenic neoantigens that were experimentally validated in a previous work[12] and detected in this study. Asterisks show epitopes that were predicted but not expressed. **d** Shared expressed neoantigens between MC38, and wild-type and RAG1$^{-/-}$ samples during progression. Neoantigens found in at least one sample from the same type are considered. **e** Shared nonsynonymous mutations and expressed neoantigens between MC38, and wild-type and RAG1$^{-/-}$ samples at day 23. Mutations and neoantigens found in at least one sample from the same type are considered. **f** Two-dimensional density plots showing the clustering of the cancer cell fractions of all mutations shared between two samples. **g** Violin plots showing tumor heterogeneity estimated from the cancer cell fractions ($n = 3$ replicates for all samples expect MC38 where $n = 1$). Box plots show the median, the 25th and 75th percentiles. **h** Number of clonal and subclonal expressed neoantigens in MC38 and all tumor samples. Error bars represent standard error of the mean ($n = 3$ replicates for all samples expect for MC38 where $n = 1$). Box plots show the median, the 25th and 75th percentiles

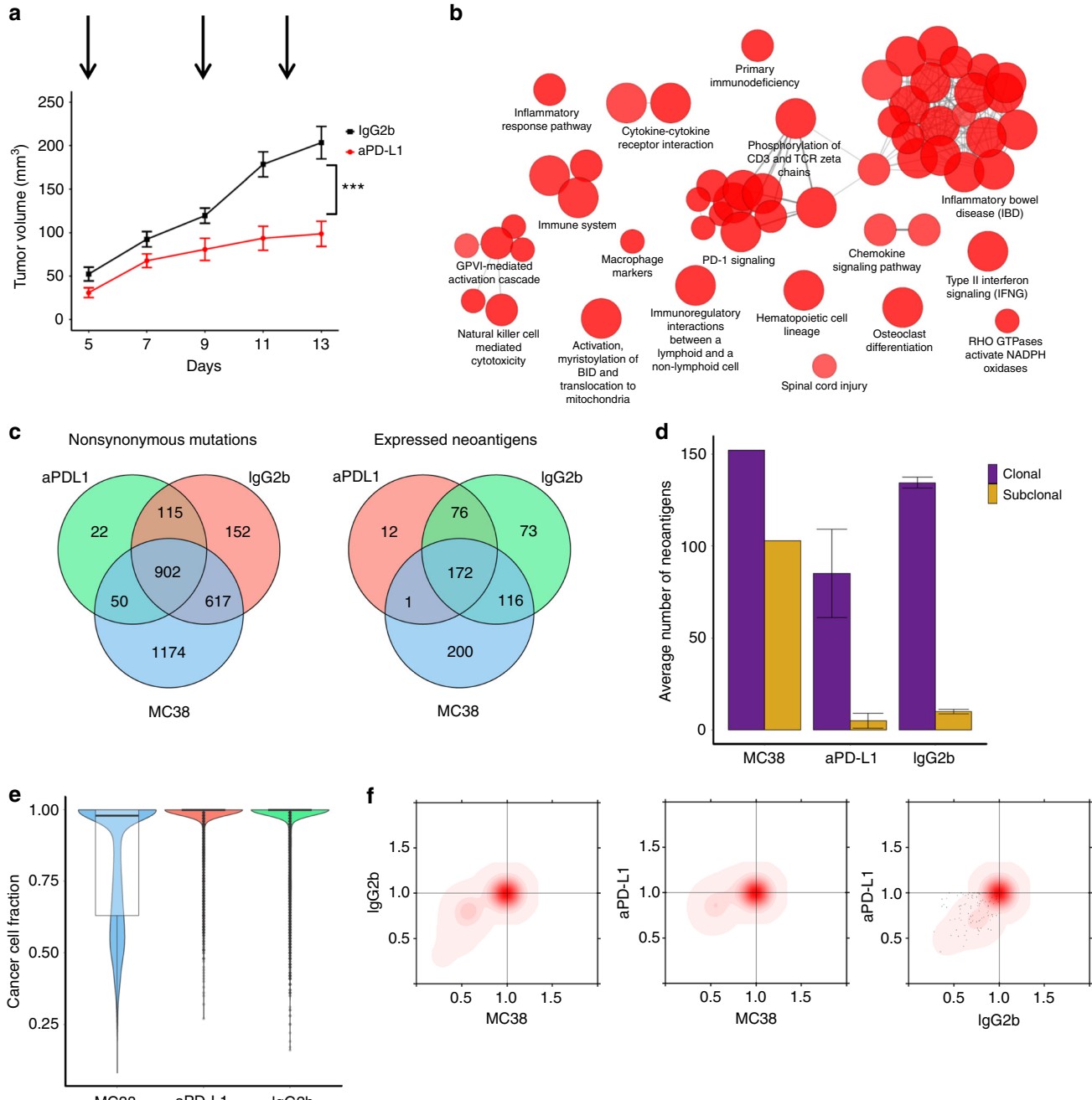

**Fig. 4** Genomic and immunogenomic impact of targeting the PD-1/PD-L1 axis. **a** Tumor growth curve in C57Bl/6 wild-type mice ($n = 11$) inoculated with $5 \times 10^5$ MC38 cells and administered with 0.5 mg of an anti-mouse PD-L1 blocking antibody as immune checkpoint inhibitor is compared with tumor growth curve seen in mice ($n = 12$) injected with IgG2b control. The data are expressed as the mean ± SEM, analyzed by a two-tailed unpaired Student's t-test. Statistical significance is indicated as *$p < 0.05$, ** $p < 0.01$, *** $p < 0.001$. **b** Enriched functions and pathways of the significantly differentially expressed genes in tumors of the anti-PD-L1 therapy vs the IgG2b control samples. The network is created using ClueGO. The pathways are functionally grouped based on the kappa score and the most significant term of each group is highlighted. The size of the nodes shows the enrichment significance after Bonferroni correction. **c** Shared nonsynonymous mutations and expressed neoantigens between MC38, and anti-PD-L1 treated and control samples. Mutations/neoantigens found in at least one sample from the same type are considered. **d** Number of clonal and subclonal expressed neoantigens in MC38 and all tumor samples. Error bars represent standard error of the mean. **e** Violin plots showing tumor heterogeneity estimated from the cancer cell fractions. **f** Two-dimensional density plots showing the clustering of the cancer cell fractions of all mutations shared between two samples; increasing intensity of red indicates the location of a high posterior probability of a cluster

L1 pathway induces very strong adaptive, and to a lesser extent innate, -mediated antitumor activity in this mouse model.

Analysis of the exome-sequencing data showed 902 mutations that were shared in all samples and 617 mutations that were detectable in the control sample and in the MC38 cell line, but absent from the anti-PD-L1-treated samples (Fig. 4c). These

mutations are potentially targeted by the immune system following blockade of the PD-1/PD-L1 pathway. A smaller number of mutations were detectable only in the anti-PD-L1-treated samples and the MC38 cell line (50). Overall, in the anti-PD-L1-treated samples the fraction of mutations resulting in expressed antigens was similar to the control sample (~25%). The

ratio of expressed neoantigens and total number of mutations was highest in the anti-PD-L1-treated sample (Supplementary Figure 13). Analysis of the peptides did not show any obvious pattern that could pinpoint rules defining the immunogenicity of the mutations (Supplementary Data 3).

A major shift was observed in the fraction of expressed neoantigens of clonal origin in both, anti-PD-L1 and control treated samples (Fig. 4d and Figure 13). The fraction of clonal neoantigens was 60, 95, and 93% in the MC38, anti-PD-L1 treated, and the control tumors, respectively. Tumor heterogeneity analysis revealed more homogenous tumors undergoing treatment with checkpoint blockers compared with the control tumors and the MC38 cell line (Fig. 4e). The same pattern can be observed in the 2D density plots, which show a shift of subclonal mutations in MC38 toward clonality in the anti-PD-L1 samples (Fig. 4f), suggesting negative selection of immunogenic subclones.

In addition, we investigated the effect of targeting the PD-1/PD-L1 axis on the cancer genome using another widely used CRC cell line, CT26. Previous genomic characterization of this cell line showed mutation in KRAS and lack of mutations in MMR, POLD1/POLE, and BRAF genes, suggesting that this cell line is a better model for non-hypermutated/MSS human CRC tumors[22]. From the 1172 point mutations in expressed genes in the CT26 cell line, 154 were in epitopes predicted to strongly bind to MHC molecules[22], showing that the neoantigen burden is about 73% lower compared with MC38 cell line.

Treatment with anti-PD-L1 antibodies reduced tumor growth in the treated mice by 50% compared with the controls (Supplementary Figure 14a), indicating that this model is less sensitive to immunotherapy with PD-L1 blockers compared with the MC38 cell line. Our results are in line with a recent study showing that tumor growth inhibition following treatment with anti-PD-L1 blocking antibodies was twice as efficient in mice transplanted with MC38 cells compared with mice transplanted with CT26 cells[31]. Analysis of the exome-sequencing data showed a large fraction of mutations that were shared in all samples (848) and 94 mutations that were not detectable in the anti-PD-L1-treated samples and therefore likely to have been targeted by the immune system (Supplementary Figure 15b). A similar number of mutations was detectable only in the anti-PD-L1-treated samples and the CT26 cell line (60) Thus, the fraction of immunoedited mutations compared with all mutations in the CT26 model was sixfold smaller than in the MC38 model, confirming that the CT26 cell line is less immunogenic than the MC38 model.

Similar to the MC38 cell line, there was a decrease in the number of neoantigens from subclonal origin in the anti-PD-L1-treated sample, albeit less pronounced (Supplementary Figure 14c, 15). The ratio of neoantigens and total number of mutations was slightly higher in the transplanted samples (Supplementary Figure 15d). There was no overlap of the peptides between the CT26 (Supplementary Data 4) and the MC38 model. The tumor heterogeneity of the anti-PD-L1 and the control samples were comparable (Supplementary Figure 14e) as would be expected by the similar numbers of targeted mutations (33 vs. 41). Finally, there was no detectable shift of subclonal mutations in CT26 toward clonality in the anti-PD-L1 samples.

Overall, the analyses of this experimental data suggest that targeting the PD-1/PD-L1 pathway potentiates immunoediting and changes the evolutionary dynamics from neutral to non-neutral in the MC38 model of hypermutated/MSI CRC. Moreover, this immunotherapeutic intervention renders the tumors more homogeneous, which could possibly explain the development of resistance to checkpoint blockers. The immunoediting effects were less pronounced in the CT26 model, likely owing to the less-immunogenic nature of this model.

**Immunoediting and acquired resistance to PD-1 blockade.** In order to test the relevance of our findings in human cancer, we analyzed genomic data from a recent study of acquired resistance to PD-1 blockade in melanoma[39]. In this work, pretreatment and relapse samples from four patients with metastatic melanoma, which were subjected to anti PD-1 blockade therapy, were analyzed by exome sequencing. Sequencing data showed that two of the tumors developed loss-of-function mutations in JAK1 and JAK2, respectively, which resulted in lack of response to IFNγ. The third tumor had a mutation in the antigen-presenting protein β2M, which prevented the immune system from recognizing the tumor, whereas the fourth tumor had no defined mutations, which could be associated with the relapse[39].

Using exome-sequencing data, we analyzed the samples taken before therapy and after relapse with respect to the changes in the mutational landscape, the tumor heterogeneity and the clonal architecture. As can be seen in Fig. 5a, a large fraction of the mutations was detectable in baseline samples and in the relapse samples in all four cases, implicating that the bulk of the mutations were not efficiently targeted. Newly generated mutations ranged between 5% (case 1) and 33% (case 2). Mutations that were potentially immunoedited following PD-1 blockade, i.e., mutations detectable only in the baseline samples ranged between 4% (case 2) and 58% (case 3). Specifically, case 3 appeared to have strong immunoediting effects on the cancer genome.

With respect to the tumor heterogeneity, targeting the PD-1/PD-L1 pathway showed a similar trend: relapsed tumors that acquired larger number of mutations became more heterogeneous (case 2 and case 4), whereas the tumor with lower number of acquired mutations became more homogeneous (case 3) (Fig. 5b). The analysis for case 1 did not reveal changes in the tumor heterogeneity likely owing to the high number of mutations in both, baseline and relapse sample (1045). Thus, in this case the impact of newly generated mutations on the tumor heterogeneity is rather small. The analyses of the clonal architecture revealed that in all tumors there was a loss of clonal mutations in the relapsed samples compared with the baseline, ranging from 1% (Case 2) to 24% (Case 3) (Fig. 5c). Tumors that became more heterogeneous had an increased number of subclonal mutations compared with the baseline (case 2 and case 4). In accordance with the immunoediting hypothesis, the relapsed sample showing a strong immunoediting effect (case 3) had the largest decrease of both, clonal and subclonal mutations, and hence, was more homogeneous.

Overall, these results indicate that immunoediting can be associated with acquired resistance to PD-1 blockade in melanoma in some tumors with specific mutational phenotypes. Targeting the PD-1 pathways in these phenotypes seems to broaden the T-cell repertoire in a way that both, clonal and subclonal mutations, are targeted and subsequently render the tumor more homogeneous. Hence, a clone that is resistant to immune attack will ultimately dominate the population. However, given the small number of cases and the variability of the results, further studies will be necessary to investigate the effects of the checkpoint blockade on the tumor heterogeneity in relapsed tumors and confirm our findings.

## Discussion
With the development of immunotherapies with checkpoint blockers as well as other immunotherapeutic strategies, including therapeutic vaccines and engineered T cells[40], the interaction of the tumor and the immune system, and the question of how the cancer genome is edited came into focus. Our understanding of the process of cancer immunoediting and its relevance for therapeutic intervention is still incomplete and requires

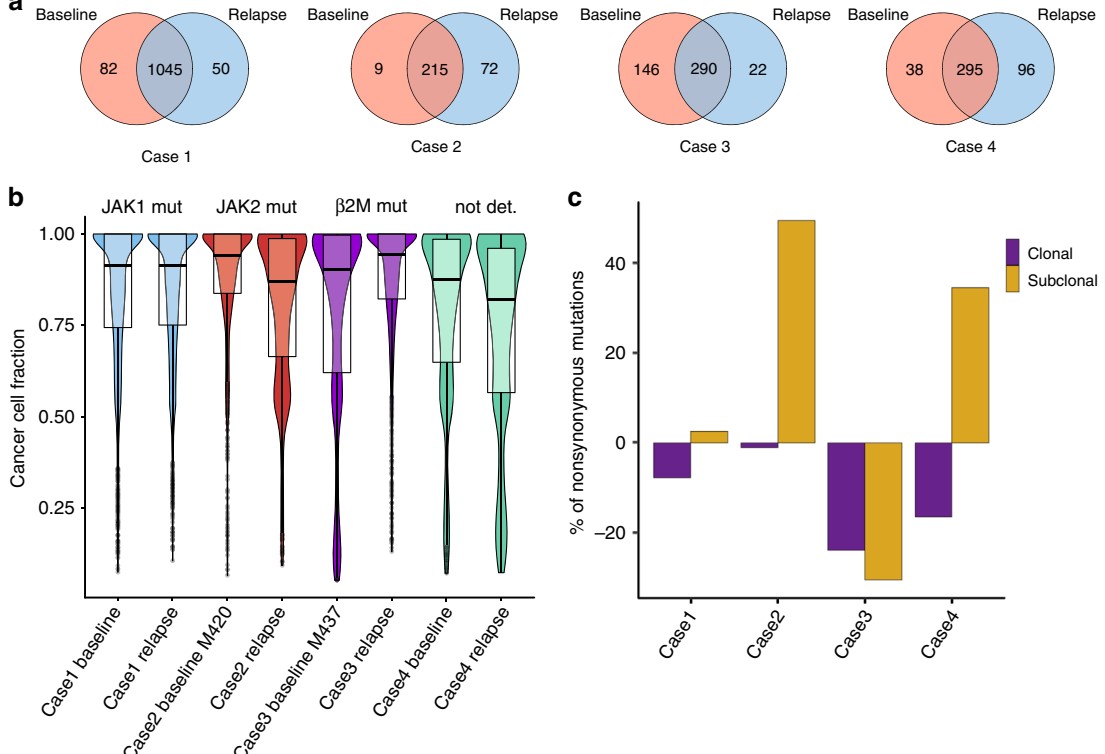

**Fig. 5** Immunoediting and acquired resistance to PD-1 blockade in melanoma. **a** Number of mutations at onset of therapy and after relapse from a study by Zaretsky et al.[39]. **b** Violin plots showing tumor heterogeneity estimated from the cancer cell fractions. Box plots show the median, the 25th and 75th percentiles. **c** Relative variation in the number of nonsynonymous mutations detected in the relapse samples compared with the baseline in the study by Zaretsky et al.[39]

comprehensive genomic analyses of longitudinal samples. Here we characterized, for the first time, the extent of immunoediting that tumors undergo during progression or as a consequence of the targeting the PD-1/PD-L1 axis. The quantification of cancer immunoediting using a mouse model of a common cancer suggests several biological conclusions and has also important implications for clinical translation.

First, neutral evolution outweighs the effects of T-cell-dependent and T-cell-independent immunoselection on the cancer genome during tumor progression in the MC38 model of hypermutated/MSI CRC tumors. Neutral tumor evolution was only recently identified, using a theoretical model that determines the expected distribution of subclonal mutations, and implies that a large number of new mutations are generated in ever smaller subclones, resulting in many passenger mutations that are responsible for intratumoral heterogeneity, but have minimal or no impact on tumor expansion[34]. In this neutral evolution model all the mutations responsible for expansion are present in the founding cell and subsequent mutations are neutral. Analysis of the TCGA data showed that CRC and some other cancers were dominated by neutral evolution whereas other cancers were not[34]. It should be noted that there are three major evolutionary forces that influence tumor progression and can shape the clonal trajectory of the tumors in this experimental setting: (1) drift owing to sampling bias of the transplants, (2) positive selection for engraftment, and (3) negative selection by the immune system. Drift can be due to random sampling of few related clones from the initial population, and would have the effect of increasing the number of the clonal composition of the transplanted tumors. The same effect would be observed in the scenario of positive selection for engraftment because of selection of clones that are able to survive transplantation. Our analyses show that the

majority of the new mutations in the transplanted tumors, both in wild-type and immunodeficient mice are subclonal, suggesting that the effects of random drift and selection due to transplantation are negligible. Last, we provide three lines of evidence that the contribution of the negative selection by the immune system is small: (1) the ratio between the expressed neoantigens and the total number of mutations did not change between different samples; (2) although the number of LOH events was increased in the transplanted tumors (both in wild-type and in immunodeficient mice), there were no LOH events at the genomic positions of the neoantigen-deriving mutations, suggesting that no neoantigens are lost due to LOH; and (3) there was no loss of expression of the shared neoantigens. Collectively, these data support the neutral evolution model in the MC38 model of hypermutated/MSI CRC tumors.

Second, targeting the PD-1/PD-L1 pathway effectively potentiates immunoediting and changes the dynamics of the system from neutral to non-neutral in the MC38 model. Currently, we can only speculate on the underlying mechanisms driving the strong immune response. It has been previously shown that immunotherapy with anti-CTLA-4 antibodies leads to a significant number of newly detected T-cell responses[41], which can be assigned to broadening of the T-cell receptor (TCR) repertoire[42]. Our data support this model in terms of therapeutic strategy blocking the PD-1/PD-L1 axis. The broadening of the TCR repertoire might be one of the mechanisms of action of anti-PD-1 treatment and could explain the success of immunotherapy in a number of malignancies. As CTLA-4 and PD-1 have differing immunological effects on circulating T cells, further mouse and human studies are necessary in order to test the hypothesis that the expansion of the TCR repertoire is a mechanism that potentiates immunoediting also in a therapeutic strategy that

blocks the PD-1/PD-L1 axis. Notably, targeting the PD-1/PD-L1 pathway in a less-immunogenic model, CT26, resulted in fivefold smaller immunoediting, and consequently less-pronounced effects on the cancer genome. Intriguingly, similar genotype-immune response associations are observed in humans CRC tumors: MSI tumors respond to checkpoint blockade, whereas MSS are refractory[28]. Further studies are necessary to investigate these genotype-immunophenotype relationships and pinpoint genetic drivers of immunoediting, and ultimately provide possible explanation for the resistance to immune checkpoint blockers in MSS patients despite the fact that tumor-infiltrating lymphocytes represent a strong independent predictor of relapse and survival[7].

And third, targeting the PD-1/PD-L1 pathway renders the tumors more homogeneous in the MC38 model. Although we did not carry out long-term experiments with different dosages and treatment schedules, one implication of this result is that the tumors might eventually become resistant to immunotherapy. We also provide data from a human study showing that in some cases tumors that relapse after PD-1 blockade are more homogeneous. Hence, cancer immunoediting could represents one mechanism of acquired immunotherapy resistance in specific mutational phenotypes. However, other mechanisms like epitope spreading[43] (immune responses to secondary epitopes) and immunodominance[44] (dominant epitopes can mask subdominant ones) could counterbalance or ameliorate this effect and thereby determine whether the tumor is eradicated or not.

On the cautionary side, the model we have used has certain limitations since it is based on a cell line, which has been edited and it does not recapitulate evolution of the tumor as it occurs naturally. However, as shown by others[12] and in this study, the MC38 model is immunogenic and responds to treatment with immune checkpoint blockers, suggesting that the MC38 cell line has evolved and acquired mutations that can be detected by the immune system. Moreover, the resemblance of the MC38 and the CT26 models to the clinical observations of the response of CRC patients treated with immune checkpoint blockers further supports the relevance of the chosen model. More sophisticated approaches using CRISPR/Cas9 technology for introducing mutations that drive spontaneous rejection of the tumor could provide additional insights into the complex evolutionary dynamics and the interaction with the immune system.

Our findings have important implications for basic research studies on mechanisms of resistance to checkpoint blockade and for clinical translation. Most importantly, given that neutral evolution, T-cell-dependent immunoediting, and T-cell-independent immunoediting are sculpting the tumor, it is of utmost importance to carry out comprehensive genomic and immunogenomic analyses of pre- and post-treatment samples. As conventional cancer therapy as well as cancer immunotherapy are altering the genomic landscape, clones that are resistant to therapy might arise and outcompete other clones. Thus, it is an imperative to characterize the applied mouse models and the evolutionary forces driving the tumor in order to dissect the contribution of individual components on shaping the cancer genome. Over and above, since MMR deficiency predicts response to checkpoint blockade in 12 different solid tumor types[45], further studies using genetically engineered mouse models[46] could help to identify molecular determinants that make not only CRC tumors sensitive to immune checkpoint blockade, but possibly also other tumor types.

Finally, our results have important implications also for clinical research. Given the fact that some cancers including CRC, stomach, lung, and bladder are dominated by neutral evolution[34], it will be important to study tumors over time to determine the impact of the immunological selection following checkpoint blockade. Theoretically, neutral evolution generates greater tumor heterogeneity and hence, may facilitate adaptation after the initiation of immunotherapy. However, investigating evolutionary dynamics within human cancer is challenging since longitudinal observations are unfeasible and both, the genetic and immune landscape of cancer, are highly dynamic and interwoven[8]. Use of new technologies such as single-cell sequencing, as well as multiregion sequencing and higher sequencing depth together with improved computational methods, as recently shown[47], will provide better understanding of the relationship between the clonal architecture of a tumor and the antitumor response of the immune system. In this context, advances in organoid and gene-editing technologies will open new avenues of research and ultimately lead to the development of effective strategies for precision immuno-oncology.

In summary, we demonstrated that neutral evolution is the major force sculpting the tumor during progression and that checkpoint blockade effectively enforces T-cell-dependent immunoselective pressure in mouse models of CRC. Our study investigates another layer of complexity of the tumor evolution and the dynamic nature of clonal selection driven by immunological and non-immunological mechanisms. An improved understanding of how the immune system affects tumor progression will be fundamental to improving response to immunotherapies and combating resistance, but will require comprehensive genomic and immunogenomic analyses of both mouse models and human samples.

## Methods

**Cell culture.** The MC38 cell line, derived from methylcholanthrene-induced C57BL6 murine colon adenocarcinoma cells was maintained in DMEM supplemented with 10% FBS, 1mM L-Glutamine, and Penicillin (100 μ/ml)–Streptomycin (0.1 mg/ml), at 37 °C under 5% CO2 pressure. The CT26.WT cell line, derived from N-nitroso-N-methylurethane-induced BALB/c (H-2d) undifferentiated colon carcinoma was maintained in RPMI1640 supplemented with 10% FBS, 1mM L-Glutamine, and Penicillin (100 μ/ml)–Streptomycin (0.1 mg/ml), at 37 °C under 5% CO2 pressure. The MC38 cell line was kindly provided by Maximilian Waldner, University of Erlangen, Germany. The CT26 WT cell line was provided by TRON. Cell lines were tested negative for mycoplasma (GATC, Konstanz, Germany). All media components were obtained from Sigma-Aldrich. For implantation cells were washed with PBS (Sigma-Aldrich), mildly trypsinized, further washed in PBS, and checked for viability before finally being dissolved in PBS at the desired densities.

**Mouse experiments.** Wild-type C57BL/6N mice $RAG1^{-/-}$ (B6.129S7-$RAG1^{tm1-Mom}$/J) mice were purchased from Charles River. Mice were maintained under SPF conditions. All animal experiments were performed in accordance with the Austrian "Tierversuchsgesetz" (BGBI. Nr.501/1989 i.d.g.F. and BMWF-66.011/0061-II/3b/2013) and were approved by the "Bundesministerium für Wissenschaft und Forschung" (bm:wf).

A total of $5 \times 10^4$ MC38 colon carcinoma cells were injected subcutaneously (s. c.) into the left flank of 8- to 12-week-old female wild-type or $RAG1^{-/-}$ mice. Tumor growth was monitored three times per week by measuring tumor length and width. Tumor volume was calculated according to the following equation: ½(length × width$^2$). Each excised tumor was randomly divided in three pieces and used for either DNA or RNA isolation or for FACS analysis. For survival analysis, mice with tumors greater than the length limit of 15 mm were killed and counted as dead.

Wild-type C57Bl/6N mice were injected s.c. with $5 \times 10^5$ MC38 cells and administered with 0.5 mg of an anti-mouse PD-L1 (Clone10F.9G2; BE0101) antibody or corresponding IgG2b (LTF-2; BE0090) control antibody (all from BioXCell, USA) i.p. every 3–4 days starting from day 1 of the MC38 challenge. Tumor growth was monitored as described above. DNA and RNA isolations were done from complete excised tumors.

Wild-type BALB/c mice were injected s.c. with $5 \times 10^5$ CT cells and administered with 0.5 mg of an anti-mouse PD-L1 (Clone10F.9G2; BE0101) or corresponding IgG2b (LTF-2; BE0090) control antibody (all from BioXCell, USA) every 3–4 days starting from day 1 of CT26 challenge. Tumor growth was monitored as described above. DNA and RNA isolations were performed from complete excised tumors.

**Immunophenotyping.** Mononuclear infiltrating cells were isolated from both subcutaneous tumors and skin tissue at the indicated time points[48]. In brief, tumor and skin tissues from killed mice were prepared by mechanical disruption followed by digestion for 45 min with collagenase D (2.5 mg/ml; Roche, 11088858001) and

DNase I (1 mg/ml; Roche, 11284932001) at 37 °C. For skin tissue Liberase (5 mg/ml; Roche, 5401020001) was added to the above described digestion mix. Digested tissues were incubated 5 min at 37 °C with EDTA (0.5 M) to prevent DC/T-cell aggregates and mashed through a 100-µm filter and a 40-µm filter. Cells were washed, and resuspended in PBS+2% FCS.

Tumor and skin-infiltrating immune cells were incubated with FcR Block (BD Biosciences, 553142) to prevent nonspecific antibody binding before staining with appropriate surface antibodies for 30 min at 4 °C, washed with PBS+2% FCS, and used for FACS analysis. For intracellular cytokine staining, cells were stimulated with 50 ng/ml Phorbol 12,13-dibutyrate (PDBu, Sigma, P1269), 500 ng ionomycin (Sigma, I0634), and GolgiPlug (BD Biosciences, 555029) for 4–5 h. After fixation with the FoxP3 staining buffer set (eBiosciences, 00-5523) for at least 30 min at 4 °C, cells were permeabilized with the fixation/permeabilization buffer (eBiosciences, 00-5523) and incubated with FcR Block (BD Biosciences, 553142) before staining with specific cell surface or intracellular marker antibodies. Data acquisition was performed on a LSR Fortessa cell analyzer (Becton Dickinson). Data analysis was conducted using the Flowlogic software (eBioscience, version 1.6.0_35).

The following antibodies were used for flow cytometry at a concentration of 1:200 with exceptions marked in the list: CD4-V500 (BD, 560783), CD45-V500 (BD, 561487), CD8a-PerCP Cy5.5 (eBiosciences, 45-0081-82), CD3-PE (eBiosciences, 12-0031-83), CD11c-PerCP Cy5.5 (eBiosciences, 45-0114-80), CD11b-PE (BD, 557397, 1:500), CD45-APC (eBiosciences, 17-0451-81), F4/80-PE-Cy7 (BioLegend, 123113), CD49b-FITC- (eBiosciences, 11-5971-81), Foxp3-FITC (eBiosciences, 11-5773-82, 1:100), IFNγ-PE-Cy7 (eBiosciences, 25-7311-82), CD25-bv421 (BioLegend, 102034), Gr-1-APC (eBiosciences, 17-5931-81, 1:500), MHCII-bv421 (BD, 561105).

**Exome- and RNA sequencing.** Library preparations and sequencing was performed at the Innsbruck Medical University Sequencing CF according to the following procedures. Whole-exome sequencing of the tumor, skin, and MC38 cell samples was performed with exome capture using SureSelectXT Mouse All Exon capture probes (Agilent Technologies Österreich GmbH, Vienna, Austria) followed by sequencing with the Ion Proton System (Ion Torrent, Thermo Fisher Scientific). For RNA sequencing, total RNA was extracted, quality validated with the Agilent Bioanalyzer, and submitted to QuantSeq 3′ mRNA-Seq library preparation, following the manufacturer's instructions (Lexogen, Vienna Biocenter, Austria). Resulting libraries were sequenced with the Ion Proton[TM] System.

**Exome-sequencing data analysis.** The sequencing reads were preprocessed through a quality control pipeline where they were trimmed to a maximum read length of 180 base pairs in addition to trimming the first nine bases with Trimmomatic[49]. The trimmed reads were then aligned to the mm10 reference genome using bwa-mem[50]. Picard was used to clean and sort the aligned bam files and to remove duplicate reads (http://broadinstitute.github.io/picard) and GATK for indel realignment and base quality score recalibration. Somatic point mutations were identified with Mutect[51] by comparing each tumor sample with the two skin samples and taking the intersection of the mutations. Insertions/deletions were called with Strelka[52] in the same way. Mutations were filtered so that only mutations with at least 10 alternative reads were considered. The somatic mutations were annotated using the Ensembl Variant Effect Predictor tool[53]. Somatic copy number estimations were derived from the exome-sequencing data using EXCAVATOR[54] by calculating log₂ ratios between the read depth of the tumor and two germline skin samples using the "pool" mode. The estimated log₂ ratios were then segmented by their novel heterogeneous shifting level model. The copy number alterations (CNAs) identified using exome-sequencing data were concordant to those in the same samples by using Affymetrix SNP Array. LOH events were derived using VarScan2. MutationalPatterns (https://doi.org/10.1101/071761) was used to infer the contribution of published mutational signatures[19].

**SNP arrays.** Genome-wide copy number profiles of two wild-type samples (day 23 and day 46), two RAG1$^{-/-}$ samples (day 13 and day 23), all six anti-PD-L1 and IgG2b samples, MC38 and skin germline DNA were obtained using the Affymetrix Mouse Diversity Array. The genotyping analyses were carried out at Eurofins Genomics (Ebersberg, Germany) using the Affymetrix Mouse Diversity Array. The SNP arrays were processed, quantile-normalized, and median-polished using the Aroma Affymetrix CRMAv2 algorithm[55] together with 351 publically available Mouse Diversity Genotyping Array CEL files which were downloaded from the Center for Genome Dynamics at The Jackson Laboratory (http://cgd.jax.org/datasets/diversityarray/CELfiles.shtml). CNAs for each probe were computed as log₂-ratios between the probe signal intensities of each sample and the reference skin sample and then those ratios were segmented using the circular binary segmentation algorithm implemented in the R package DNAcopy[56].

**Tumor heterogeneity.** Normal contamination estimates were calculated using the homozygous point mutations in the cell line MC38. Considering that the purity of the cell line is 1, we checked the VAF of the homozygous mutations in MC38 in all the samples together with the estimated copy numbers of the corresponding region. The expected VAF of these mutations should be 1 in all samples assuming that there is no normal contamination and no new mutations appearing in the mouse

samples at the same genomic position. As an estimate of the purity of the tumor, we took the mean of the VAF of those mutations found in a diploid region. These estimates were used to correct the mutation VAFs or copy number estimates in the rest of the analyses.

The CCF of each mutation was calculated by integrating the above mentioned purity estimates and the copy numbers from EXCAVATOR, using the approach of McGranahan et al. In brief, the VAF of each mutations, given the CCF, can be calculated as follows:

$$\mathrm{VAF(CCF)} = p * \mathrm{CCF}/[\mathrm{CN_n} * (1 - p) + p * \mathrm{CN_t}]$$

where $p$ is the tumor purity, and $\mathrm{CN_t}$ and $\mathrm{CN_n}$ are the tumor and the normal locus specific copy number. The expected number of mutated reads $x$ follows a binomial distribution with a total depth of $N$, such that the probability of a given CCF can be estimated using $P(\mathrm{CCF}) = \mathrm{binom}(x|N, \mathrm{VAF(CCF)})$. CCF values can then be calculated over a uniform grid of 100 CCF values (0.01,1) and then normalized to obtain a posterior distribution. Using this approach, mutations were classified as clonal if the 95% CCF confidence interval overlapped 1, and subclonal otherwise. Subclonal clusters of mutations were identified using a previously described statistical modeling of the distribution of clonal and subclonal mutations by a Bayesian Dirichlet process[57–59].

Tumor heterogeneity of the mutational data from human melanoma patients was analyzed using the allele-specific CNAs in addition to the VAF and purity that was provided with the data. For each mutation, the observed mutation copy number, $n_{\mathrm{mut}}$ (the fraction of tumor cells carrying a given mutation multiplied by the number of chromosomal copies at that locus) was calculated as:

$$n_{\mathrm{mut}} = \mathrm{VAF}\frac{1}{p}[p\mathrm{CN}_t + \mathrm{CN}_n(1 - p)]$$

where VAF is the variant allele frequency of the mutation, $p$ is the tumor purity, and $\mathrm{CN}_t$ and $\mathrm{CN}_n$ are the tumor and the normal locus specific copy number. As mutations that are present of multiple chromosomal copies will have a mutation copy number higher than 1, we determined the number of chromosomes that the mutations is residing on. This was done so that for all mutations in amplified regions with a copy number of CNt, the observed fraction of mutated reads is compared with the expected fraction of mutated reads resulting from a mutation present on 1,2,3,…,CN_t copies, considering a binomial distribution. The CCF was then calculated as the mutation copy number divided by the value of $C$ with the maximum likelihood. Mutations were defined as clonal if the CCF was > 0.95, and subclonal otherwise.

**RNA-seq data analysis.** The sequencing reads were first preprocessed through a quality control pipeline consisting of adapter removal with Cutadapt (http://dx.doi.org/10.14806/ej.17.1.200) and quality trimming with Trimmomatic[49] to remove bases with bad quality scores and reads shorter than 22 nucleotides. The quality trimmed reads were then mapped to the mm10 reference genome using a two-step alignment method; alignment with STAR[60] followed by alignment of the unmapped reads with Bowtie2. From the reads that mapped to multiple locations in the genome only the primary alignment was retained. Reads that mapped to ribosomal RNA locations in the genome were removed from further analysis using the split_bam.py script from the quality control package RSeQC[61]. Gene-specific read counts were calculated using HTSeq-count[62]. The R package DESeq2[63] was used for differential expression analysis. The p-values were adjusted for multiple testing based on the false discovery rate using the Benjamini–Hochberg approach.

**Neoantigens and CGAs.** All possible 8–11 mer mutated peptides generated from all the nonsynonymous mutations (missense and nonsense) were used as an input to netMHCpan to predict their binding affinity to the C57BL/6 MHC class I alleles H-2K$^b$ and H-2D$^b$. Among the candidate antigenic peptides, only the strong binders with binding affinity < 500 nM, and peptides arising from expressed genes were retained for further analysis. A mutation was considered expressed if the normalized counts of the corresponding gene were >5.

The list of CGA was downloaded from the Cancer-Testis database[64]. Their expression level was estimated using the normalized counts from DESeq2.

**Statistical analysis.** For comparison of two sample groups, a two-tailed unpaired Student's t-test was performed. Analysis and visualization of Gene Ontology terms and pathways associated with differentially expressed genes was performed using ClueGO[65]. A p-value of <0.05 was considered statistically significant: *p < 0.05; **p < 0.01; ***p < 0.001.

**Data availability.** The mouse expression data and the SNP array data were deposited in the GEO under the accession number GSE93018. The exome sequencing bam files were deposited in the Sequence Read Archive under the accession number SRP095725. Mutational data from the melanoma patients[39] were provided by Dr. Antoni Ribas. The authors declare that all the other data supporting the findings of this study are available within the article and its supplementary information files and from the corresponding authors upon reasonable request.

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

## Acknowledgements

Mutational data from the melanoma patients were kindly provided by Dr. Antoni Ribas. This work was supported by the Austrian Science Fund (DK Molecular Cell Biology and Oncology, W1101-B18), the Tiroler Standortagentur (Bioinformatics Tyrol), the European Commission (Horizon2020 project APERIM: Advanced bioinformatics tools for personalized cancer immunotherapy, Project No. 633592), and the Austrian National Bank (Jubiläumsfondsprojekt No. 16534).

## Author contributions

Z.T conceived the project. M.E. analyzed the mouse data. D.R. organized and managed the data transfer and storage, and developed read processing pipeline. P.C. M.E., F.F., and H.H. analyzed the human data. ML analyzed the CT26 genome. A.K. carried out NGS. V. K. and N.H.-K. carried out the mouse experiments and the FACS analysis. M.E., D.R., G. B. A.K., and Z.T interpreted the results. M.E. and Z.T. wrote the manuscript.

## Additional information

**Competing interests:** The authors declare no competing financial interests.

