## [Peer Review File · Nature Communications]

Reviewer #1 (Remarks to the Author):

This study explores the relative importance of neutral versus immune mediated evolution of tumor heterogeneity and immunogenicity using an established colorectal cancer tumor cell line (MC38) in RAGKO versus WT mice. The authors use genomics and bioinformatics approaches to study changes in the mutational patterns of MC38 in immunodeficient and immunocompetent environments and reach a series of three conclusions. First, that neutral evolution of tumor heterogeneity outweighs the effects of immunoselection (i.e., cancer immunoediting) during tumor progression in this particular tumor model. Second, targeting the PD-1/PD-L1 pathway potentiates immunoediting. Third, targeting the PD-1/PD-L1 pathway renders tumors more homogeneous.

Whereas the general approach used in this study is very clever, its execution is somewhat flawed by the choice of the tumor cell line used.

By starting with a tumor that has already undergone cancer immunoediting because it was derived from a wild type immunocompetent mouse, the investigators have precluded seeing effects on the strong tumor neoantigens that may have been generated during the initial transformation process. It is these antigens that the natural immune system would have already selected against and thus there would be nothing for the unmanipulated immune system to go after in these experiments. Thus, their analyses for their first point is totally predictable based on their experimental design. The more relevant experiment would have been to use a tumor cell line that is derived from immunodeficient mice and then to explore what happens to expression of the neoantigens that are sufficient to drive spontaneous rejection of the tumors in WT mice. Thus, the current study cannot make conclusions about the role of immunity versus neutral evolution on mechanisms of tumor control/escape.

Along similar lines, the mutations driving tumorigenesis may be very different between different tumors or even different individual tumors of the same type. Thus, it is possible that by the time that MC38 evolved, it acquired mutations that established the "set point" of immunogenicity that could be detected by the unmanipulated immune system. Additionally, all the work presented is performed with a single tumor cell line. This raises the question of whether the results presented are generalizable to other CRC tumors or tumors of other types.

Finally, the authors use bioinformatics approaches to identify tumor neoantigens. However, it is well established that only a small percentage of predicted neoantigens are actually immunologically recognized. This study thus lacks functional confirmation of some neoepitopes that the authors identify other than the few that have been validated by others.

However, having said this, I do like the experiments that explore points two and three. This is because they now look at therapeutically-induced cancer immunoediting. These are very nice experiments that do not suffer from many of the same issues as the first set of experiments. My only concern here is the use of only one type of tumor cell line. I certainly feel that before the conclusions can be supported, the authors would need to show that results similar to that observed with MC38 are also seen with other colorectal cancer cell lines and tumors of other tissue types. Certainly, the assessment of changes in melanoma following immunotherapy helps but the systems are completely different and it is important to show that the initial results with MC38 can be recapitulated with other experimental tumors using the same experimental system.

Reviewer #2 (Remarks to the Author):

In this study, the authors present a very interesting mouse model of cancer immunoediting occurring during tumour engraftment as well as under immune checkpoint inhibitors. Given the impact of immunotherapy in cancer and the growing interest in understanding the interactions

between malignancies and the immune system, this is a very important study describing an informative model of cancer immunoediting.

Although the study is very interesting and novel, there are several major concerns that need to be addressed by the authors.

Major points:

- Rather than just looking at C>A transversion in MC38, the authors should perform a more accurate genomic signatures analysis and verify that MMR signatures are present, thus confirming MC38 as a good model for MSI+ colorectal cancer.
- Are the mutations in driver genes in MC38 in known hotspots?
- The authors describe upregulation of PD1 as the major mechanism of immune escape in tumours seeded in WT mice. Is this due to the inherent plasticity of cancer cells or due to bet-hedging, hence pre-existing variability of PD1 expression in the original MD38 population (i.e. only the high PD1 clones survived)?
- I would argue that the increase mutational load after transplantation with respect to the original MC38 line might be just due a clonal bottleneck in the transplanted population (either due to immunoediting or just selection for the mouse environment). Are indeed all the new mutations clonal or subclonal?
- CCF plots in Figure 3B & F to me don't make sense. According to the Venn diagrams (and as expected), the large majority of variants are shared between the original line and the transplanted populations. Those should be largely clonal in all samples (modulo some normal contamination in the transplanted samples), hence one should expect a large clonal cluster at CCF=1. However, the major cluster is shifted around ~60% CCF, which is odd, as at least MC38 should be a pure sample. Was this a problem of copy number adjustment? One way to test this is plotting only mutations in diploid regions and multiply the VAF times 2, to see if you get the same distribution. Indeed, if Figure 3F was correct, that would indicate that in all samples there is a large 60% CCF subclone that does not change in frequency over time and over transplantations. This could be true, or that could just be the clonal cluster that has been mis-corrected. Usually I expect a large clonal cluster, which is not evident in the current plots. It would be best also to see also the raw VAF plots, as Dirichlet clustering is often overfitting and confusing.
- Moreover, if the 60% subclone was in fact the miscorrected clonal cluster, then it would make sense its mutations stay stable over time, as clonal mutations cannot change in frequency, except the whole tumour goes extinct.
- The authors do not discuss the problem of sampling bias. Just the fact that a mutation is not picked up at a later time point does not necessarily imply it was selected against. It might be simply due to a sampling problem (intra tumour heterogeneity).
- Does each circle in Figure 3D represent a single mouse? How much variability is there between replicas?
- Figure 3A,D and E are hard to interpret in absolute numbers, the key value to understand immunoediting would be the ratio between expressed neoantigens and total nonsynonymous mutations.
- Also I feel they might over interpret the results in Figure 5A. It is a bit challenging to see a clear trend of homogenising or heterogenising effects in 4 cases of melanoma. I would be a bit more careful and critical here.
- The authors claim that that immunoediting "counterbalances" neutral evolution. I would rather say that immunoediting, as any form of selection, changes the dynamics from neutral to non-neutral.
- In general, my main critic is that, although the underlying evolutionary dynamics in this system are non-trivial, such complex dynamics are not explicitly discussed in the paper. In this experiment we have three major evolutionary forces that could change the mutational profile of a population: (A) drift, (B) positive selection for engraftment and (C) negative selection by the immune system. (A) The original MC38 population has a large number of clonal mutations and a certain number of subclonal mutations (at least those detectable with current technologies). Drift

can be due to random sampling a few related clones from the initial population, and would have the effect of increasing the number of clonal mutations in the mouse, as the transplanted clones will have, just by chance, a more recent common ancestor than the clones in the original population. Given the large size of the transplanted population, I assume this effect to be negligible. (B) Positive selection from engraftment will select only those clones that are able to survive transplantation. If the surviving clones are more related than average (hence having a more recent common ancestor), which would result in more clonal mutations in the engrafted population. If a single clonal lineage survives engraftment, the number of new clonal variants in the mouse could be very large. (C) Finally, the immune system could also select only subclones with low immunogenicity. This is the key phenomenon the authors want to study. In this case subclones with a lower ratio between expressed neoantigens and total number of mutations will be selected for. Importantly, those clones may still have a lot of mutations, just not the neoantigen generating ones. That's why the ratio is more important than the absolute numbers. Moreover, these clones may have lost neoantigen-generating mutations that were clonal in MC38. As reverting mutations is very unlikely, that could occur by losing the allele with the variant. Do the authors observe more LOH events in the transplanted populations? Finally, if all these three forces are negligible, then in a perfectly neutral scenario, then indeed all clonal mutations and detectable subclonal mutations should stay the same, although newly generated subclonal mutations at very low frequency will be present in the transplanted populations (but not detected). This corresponds to sampling a neutral phylogenetic tree multiple times through the experimental passages. Are the data consistent with this latest hypothesis? From the Venn diagrams this is unclear.

Minor comments:

- should be 'microsatellite unstable'

Reviewer #3 (Remarks to the Author):

This is a very good original work providing a comprehensive genomic, transcriptomic and immunogenomic analysis. With regard to next-generation sequencing and regulatory networks, this manuscript could be improved by including whole-genome sequencing for the identification of large structural genome changes, such as copy number alterations and chromosomal rearrangements, which play a crucial role in tumorigenesis and metastasis. Moreover, the lack of a computational systems biology approach in this study limits the understanding of non-linear transcriptional networks that control biological and intracellular systems.

We appreciate very much the comments of the referees and their suggestions. We addressed all issues raised by the referees and by doing so considerably improved the manuscript. Major changes include:

- 1) Additional experiments using another colorectal cancer cell line as requested by reviewer 1. We used the CT26 cell line, a model for non-hypermutated colorectal cancer and carried out experiments with anti-PD-L1 antibodies to investigate the effect of strong immunological pressure in this mouse model;
- 2) Additional computational analysis as requested by reviewer 2. We provide now mutational signatures, LOH data, ratios of neoantigens and mutations, and raw VAF plots; and
- 3) In-depth discussion of the complex evolutionary dynamics as requested by reviewer 2. We adapted also the conclusions and the title according to the suggestions.

Additionally, we addressed also all other points and added nine supplementary figures and two tables with the results of the analyses of the additional experiments and the computational analyses. The additional experiments as well as the computational analyses provide further support for our initial findings and answers to the poised questions.

The changes in the manuscript are marked in red and the point-by-point responses to the reviewers are in the following pages.

We hope that these revisions will satisfy the criticisms of the reviewers.

Reviewer #1 (Remarks to the Author):

“...Whereas the general approach used in this study is very clever, its execution is somewhat flawed by the choice of the tumor cell line used.

By starting with a tumor that has already undergone cancer immunoediting because it was derived from a wild type immunocompetent mouse, the investigators have precluded seeing effects on the strong tumor neoantigens that may have been generated during the initial transformation process. It is these antigens that the natural immune system would have already selected against and thus there would be nothing for the unmanipulated immune system to go after in these experiments. Thus, their analyses for their first point is totally predictable based on their experimental design. The more relevant experiment would have been to use a tumor cell line that is derived from immunodeficient mice and then to explore what happens to expression of the neoantigens that are sufficient to drive spontaneous rejection of the tumors in WT mice. Thus, the current study cannot make conclusions about the role of immunity versus neutral evolution on mechanisms of tumor control/escape

Along similar lines, the mutations driving tumorigenesis may be very different between different tumors or even different individual tumors of the same type. Thus, it is possible that by the time that MC38 evolved, it acquired mutations that established the “set point” of immunogenicity that could be detected by the unmanipulated immune system. Additionally, all the work presented is performed with a single tumor cell line. This raises the question of whether the results presented are generalizable to other CRC tumors or tumors of other types....”

We completely agree with the reviewer that the MC38 cell line model has certain limitations. The reason for selecting this cell line was three-fold.

First, it is widely used cell line for CRC with >330 publications.

Second, it is a good model for hypermutated/MSI CRC. Since MSI phenotype is responding (whereas MSS is refractory) to immune checkpoint blockade (ref. Le et al., NEJM 2015), we decided to focus on a hypermutated/MSI model.

Third, as pointed out by the reviewer although the cell line has already been edited, the MC38 is immunogenic and has been used in studies with checkpoint blockers (e.g. ref. Yadav et al, Nature 2014). It is very likely that the explanation the reviewer provided holds true, i.e. that the MC38 evolved and acquired mutations that established the “set point” of immunogenicity. As our intention was to investigate tumor progression as well as the impact of therapeutically-induced cancer immunoediting, we chose this cell line.

The results we present are not generalizable to other CRC tumors (as shown with the additional experiment with another CRC cell line), which further demonstrate the complexity of the evolutionary dynamics and the selective pressure induced to the system. Hence, it is of utmost importance to select the appropriate model system to study the questions addressed. We strongly believe that despite the limitations of the model we chose, we can make conclusions about the role of immunity on tumor progression. However, we followed reviewer’s comment and provide now a more cautionary note (page 12, third paragraph).

“...Finally, the authors use bioinformatics approaches to identify tumor neoantigens. However, it is well established that only a small percentage of predicted neoantigens are actually immunologically recognized. This study thus lacks functional confirmation of some neoepitopes that the authors identify other than the few that have been validated by others...”

We agree with the reviewer that these neoantigens were predicted in silico and only a fraction of them might be immunologically recognized. As of today, computational prediction of the immunogenicity of neoantigens is not possible. Furthermore, experimental identification of neoepitopes targeted by T cells is very difficult and not scalable, as evident by a handful of reported neoantigens that are recognized by T cells. Despite these limitations, we believe that this type of analysis is valuable as shown by us and other researchers (e.g. Ref. 6 Rooney et al, Cell 2015).

“...However, having said this, I do like the experiments that explore points two and three. This is because they now look at therapeutically-induced cancer immunoediting. These are very nice experiments that do not suffer from many of the same issues as the first set of experiments. My only concern here is the use of only one type of tumor cell line. I certainly feel that before the conclusions can be supported, the authors would need to show that results similar to that observed with MC38 are also seen with other colorectal cancer cell lines and tumors of other tissue types. Certainly, the assessment of changes in melanoma following immunotherapy helps but the systems are completely different and it is important to show that the initial results with MC38 can be recapitulated with other experimental tumors using the same experimental system...”

We appreciate very much this comment of the reviewer. We completely agree that investigating the impact of therapeutically-induced cancer immunoediting is highly relevant. We also followed the recommendation and carried out additional experiments using the colorectal cancer cell line CT26. This cell line is a model for non-hypermutated/MSI- human CRC and is less immunogenic than MC38 (page 9 and supplementary figure S14). The results show that the fraction of the immunoedited mutations was fivefold smaller than in the MC38 model. Intriguingly, the results of the analyses with the MC38 and CT26 resemble clinical observations in MSI+ and MSI- patients on immunotherapy, further supporting the relevance of the chosen model (page 12, first paragraph and third paragraph)

Reviewer #2 (Remarks to the Author):

“...Rather than just looking at C>A transversion in MC38, the authors should perform a more accurate genomic signatures analysis and verify that MMR signatures are present, thus confirming MC38 as a good model for MSI+ colorectal cancer...”

We thank the reviewer for this very useful suggestion. We carried out additional analysis as suggested and report the results in the revised manuscript (page 5, first paragraph and Supplementary Figure S1). While the MC38 cell line shows a similar but distinct pattern than found in spontaneous primary human CRC (likely due to the treatment with dimethylhydrazine), we advocate that it is a valid model for hypermutated and/or MSI CRC due to: 1) existence of MMR signature, 2) prevalence of C>A mutations, 3) mutations in MMR and POLD1 genes, and 4) mutation in BRAF gene. We now also report the results with another cell line CT26, which is a better model for non-hypermutated tumors (no MMR mutations, no POLD1/POLE mutations, KRAS mutation).

“...Are the mutations in driver genes in MC38 in known hotspots?...”

We appreciate the comment and carried out the hotspot analyses which we now report on page 5, second paragraph and Supplementary Table 1.

“...The authors describe upregulation of PD1 as the major mechanism of immune escape in tumours seeded in WT mice. Is this due to the inherent plasticity of cancer cells or due to bet-hedging, hence pre-existing variability of PD1 expression in the original MD38 population (i.e. only the high PD1 clones survived)?...”

This is an important question and we included the answer on page 6: “Although we cannot exclude bet-hedging (i.e. only clones with higher PD-L1 expression survive) due to the lack of single-cell sequencing data, our finding is in accordance with previous studies showing that PD-L1 expression of MC38 transplanted tumors increases as a result of exposure to inflammatory cytokines such as INF-gamma, which is sufficient for tumor escape and immune evasion”.

“...I would argue that the increase mutational load after transplantation with respect to the original MC38 line might be just due a clonal bottleneck in the transplanted population (either due to immunoediting or just selection for the mouse environment). Are indeed all the new mutations clonal or subclonal?...”

Most of the newly acquired mutations in the transplanted tumors (both in the wild type and the immunodeficient mice) are subclonal. Additional information regarding this is added to the supplementary materials (Figure S7 and S8).

“...CCF plots in Figure 3B & F to me don't make sense. According to the Venn diagrams (and as expected), the large majority of variants are shared between the original line and the transplanted populations. Those should be largely clonal in all samples (modulo some normal contamination in the transplanted samples), hence one should expect a large clonal cluster at CCF=1. However, the major cluster is shifted around ~60% CCF, which is odd, as at least MC38 should be a pure sample. Was this a problem of copy number adjustment? One way to test this is plotting only mutations in diploid regions and multiply the VAF times 2, to see if you get the same distribution. Indeed, if Figure 3F was correct, that would indicate that in all samples there is a large 60% CCF subclone that does not change in frequency over time and over transplantations. This could be true, or that could just be the clonal cluster that has

been mis-corrected. Usually I expect a large clonal cluster, which is not evident in the current plots. It would be best also to see also the raw VAF plots, as Dirichlet clustering is often overfitting and confusing. • Moreover, if the 60% subclone was in fact the miscorrected clonal cluster, then it would make sense its mutations stay stable over time, as clonal mutations cannot change in frequency, except the whole tumour goes extinct... ”

We agree with the reviewer’s observations that this is an unusual clonal composition and have taken these concerns into account and report now also raw VAF plots. As can be seen from the VAF density plots of the mutations found in diploid regions (both raw VAF and VAF multiplied by 2 are included in the supplementary materials (Figure S9)), there is a large number of potentially subclonal mutations with VAF ~ 30%. One possible explanation is that the cell line is highly heterogeneous with a complex branching clonal trajectory. Another possibility is that there are many subclonal CNAs in which case the estimated CCF would have to be adjusted additionally. In order to take into account the possibility of subclonal CNAs in our heterogeneity analysis, we would also need to obtain allele-specific CNAs and for this we need heterozygous germline mutations. However, these are samples from inbred mice which do not have many heterozygous germline mutations and therefore further analysis cannot be done.

“...The authors do not discuss the problem of sampling bias. Just the fact that a mutation is not picked up at a later time point does not necessarily imply it was selected against. It might be simply due to a sampling problem (intra tumour heterogeneity)... ”

We now discuss the matter and included a comment on page 7, second paragraph and page 14, mouse experiments. Given the similarity of the biological replicates, ie. number of shared mutations between all the samples (Figure S7), as well as their similar CCFs we concluded that the sampling bias is negligible.

In the second experimental setup (α -PDL1 vs IgG), DNA and RNA isolations were done from complete excised tumors (comment included on page 14, mouse experiments).

“...Does each circle in Figure 3D represent a single mouse? How much variability is there between replicas?... ”

Each circle in Figure 3D represents union of the expressed neoantigens from three replicates. Around 70-80% of the neoantigens are shared between the three replicates that represent one circle, however there is some variability and for this reason we have used a union of all three replicates. For better overview, we included sample-specific venn diagrams in the supplementary materials (Figure S7, S8 and S15).

“...Figure 3A,D and E are hard to interpret in absolute numbers, the key value to understand immunoediting would be the ratio between expressed neoantigens and total nonsynonymous mutations... ”

We thank the reviewer for this important suggestion. We included the ratio between expressed neoantigens and total nonsynonymous mutations in the supplementary materials (page 7 last paragraph, Figure S5, S12 and S14). As can be seen from the plots, the ratio is similar between samples (Figure S5) and changes after treatment (Figure S12).

“...Also I feel they might over interpret the results in Figure 5A. It is a bit challenging to see a clear trend of homogenising or heterogenising effects in 4 cases of melanoma. I would be a bit more careful and critical here... ”

We appreciate the comment and included a more cautionary note about the interpretation of the results in Figure 5A (page 10, last paragraph).

“...The authors claim that that immunoediting “counterbalances” neutral evolution. I would rather say that immunoediting, as any form of selection, changes the dynamics from neutral to non-neutral...”

We thank the reviewer for this comment and we have modified the title and the manuscript accordingly (page 9, last paragraph).

“...In general, my main critic is that, although the underlying evolutionary dynamics in this system are non-trivial, such complex dynamics are not explicitly discussed in the paper. In this experiment we have three major evolutionary forces that could change the mutational profile of a population: (A) drift, (B) positive selection for engraftment and (C) negative selection by the immune system.

(A) The original MC38 population has a large number of clonal mutations and a certain number of subclonal mutations (at least those detectable with current technologies). Drift can be due to random sampling a few related clones from the initial population, and would have the effect of increasing the number of clonal mutations in the mouse, as the transplanted clones will have, just by chance, a more recent common ancestor than the clones in the original population. Given the large size of the transplanted population, I assume this effect to be negligible.

(B) Positive selection from engraftment will select only those clones that are able to survive transplantation. If the surviving clones are more related than average (hence having a more recent common ancestor), which would result in more clonal mutations in the engrafted population. If a single clonal lineage survives engraftment, the number of new clonal variants in the mouse could be very large.

(C) Finally, the immune system could also select only subclones with low immunogenicity. This is the key phenomenon the authors want to study. In this case subclones with a lower ratio between expressed neoantigens and total number of mutations will be selected for. Importantly, those clones may still have a lot of mutations, just not the neoantigen generating ones. That’s why the ratio is more important than the absolute numbers. Moreover, these clones may have lost neoantigen-generating mutations that were clonal in MC38. As reverting mutations is very unlikely, that could occur by losing the allele with the variant. Do the authors observe more LOH events in the transplanted populations?

Finally, if all these three forces are negligible, then in a perfectly neutral scenario, then indeed all clonal mutations and detectable subclonal mutations should stay the same, although newly generated subclonal mutations at very low frequency will be present in the transplanted populations (but not detected). This corresponds to sampling a neutral phylogenetic tree multiple times through the experimental passages. Are the data consistent with this latest hypothesis? From the Venn diagrams this is unclear...”

We appreciate very much the important issues pointed out by the reviewer. The manuscript has been modified and now discusses the underlying evolutionary dynamics of the system in more detail (page 11, second paragraph). As previously mentioned, the newly acquired mutations in the transplanted tumors are mostly subclonal which indicates that scenario A and B are not the evolutionary forces that shape the tumor progression. As for scenario C, which is the key phenomenon that we are interested in, we have several indications that this is not the case in our first experiment. Firstly, the ratio between the expressed neoantigens and the total number of mutations is not changed between different samples and the cell line (page 7 last paragraph and Supplementary Figure S5). Moreover, even though we have identified increased number of LOH events in the transplanted tumors (both in the wild type and in the immunodeficient mice), there were no LOH events at the neoantigens genomic positions, suggesting that no neoantigens are lost due to loss of heterozygosity (page 7, last paragraph). Finally, we did not observe loss of expression of the shared neoantigens. All of these observations, together with the fact that most of the clonal and subclonal mutations stay the same, led us to the conclusion that this tumor progresses in a neutral scenario unless exposed to a stronger selection pressure such as immunotherapy (page 11, second paragraph).

Reviewer #3 (Remarks to the Author):

“...This is a very good original work providing a comprehensive genomic, transcriptomic and immunogenomic analysis. With regard to next-generation sequencing and regulatory networks, this manuscript could be improved by including whole-genome sequencing for the identification of large structural genome changes, such as copy number alterations and chromosomal rearrangements, which play a crucial role in tumorigenesis and metastasis. Moreover, the lack of a computational systems biology approach in this study limits the understanding of non-linear transcriptional networks that control biological and intracellular systems...”

We thank the reviewer for his comments and suggestions. We agree that the manuscript would have been improved with whole-genome sequencing which would enable us to identify chromosomal rearrangements and it is preferable for copy number alterations analysis. However, whole-genome sequencing provides less coverage which is of utmost importance for the heterogeneity analysis. Moreover, we also provide copy number alterations identified using Affymetrix SNP Array. Hence, we strongly believe that the data set we generated was highly appropriate for the questions addressed.

We also agree with the reviewer that computational systems biology approaches would enhance our understanding of the underlying networks. However, given the small number of samples prevents the application of more sophisticated systems biology techniques. Our focus here was the investigation of the effects of immunoediting on tumor progression and how is immunotherapy modulating immunoediting. Our results point directions for further studies aiming at reconstructing networks using systems biology approaches.

The authors have done an excellent job responding to the criticisms. For the most part, I now find the manuscript ready for publication.

There is one additional point however that the authors need to address. In the original description of "Cancer Immunoediting" the originators of the concept stated that "editing" could be a consequence of selection for tumor variants that did not express particular antigens but also noted that "editing" could also induce changes in the tumor microenvironment or changes in the tumor cells themselves that reduced their immunogenicity independently of the presence or absence of antigens. This latter concept is, in fact, what the current paper nicely shows. That is, that expression of immune checkpoints on tumor cells renders the cells capable of growing in an immunologically competent host and thus requires the presence of checkpoint blocking antibodies to achieve a selection for tumor cells that are still capable of growing out. Thus I do think the authors should emphasize the fact that their work now helps to correct a common misconception about cancer immunoediting and supports the concept as originally proposed.

Reviewer #2 (Remarks to the Author):

Although the authors did attempt to address my comments, there are quite a few important points of concern that need to be addressed, especially regarding the new data presented on the CT26 cell line:

1) page 9

here the authors present a new analysis using the same approach but on a different cell line (CT26). What I find completely puzzling, is that the new cell line has the same subclonal composition than MC38, with a large subclonal cluster at around 70% CCF. This cannot possibly be right, if I was dubious of the author's explanation of a complex subclonal architecture and branching structure in MC38, now I am convinced there must be something wrong in the variant calling as such complex structure cannot be the same also in CT26, particularly because the individual variants are completely different. Maybe there is something wrong in the reference sample used. The authors must address this problem!

2) row 311-313

the ratio of expressed neoantigens and total number of exonic mutations should DECREASE under immunoediting because the immune system gets rid of mutations generating neoantigens, but not the others...

3) row 316

why in the anti-PDL1 there are MORE clonal expressed neoantigens? Immunoediting would predict there should be less

4) row 321

if the hypothesis of immunoediting is correct, then the CCF shift should be due to negative selection for a lot of immunogenic subclones and consequently the rise in frequency of the non-immunogenic ones, hence this is rather negative selection, not positive (Darwinian) selection.

5) in the abstract, rows 47-54

here do the authors just mean that when tumours are hiding from the immune system (e.g. through upregulation of PD1), then immunoediting is weak and the neutral accumulation of mutations dominates? After this initial sentence, do they mean immunoediting start playing a role when PD1 is suppressed? If this is the case, they should spell it out more clearly in the abstract as I struggled to get it.

I would also avoid the general statement that "neutral evolution is THE major force sculpting the tumour" as even in the case of a neutrally evolving tumour, the past there must have selection for driver alterations.

6) rows 195-199

the authors seem to suggest that the upregulation of PD1 occurs without a selective sweep for high-PD1 clones, but rather through plasticity of cells switching to a high-PD1 program. Here I would avoid explicitly refer to bet-hedging as it would not resonate with the general readership and just state that the upregulation of PD1 is likely due to cell phenotypic plasticity, not positive selection for high-PD1 clones.

7) figure 3A

the venn diagrams are not weighted and do not convey the message that most mutations are in common to all three cases

8) row 224

can please the authors refer to a venn diagram to illustrate the mutations shared between the replicas?

9) row 227-230

I don't understand this sentence at all

10) row 283-284

not sure what the authors mean here, it seems they simply do not observe positive (Darwinian) selection because the subclonal architecture does not change over time

11) row 420

in the discussion the authors reported my comments on the driving evolutionary forces at play here, they should note that this refers to their experimental setting, not generally to evolution of human tumours, where there is no engraftment of course, and there could be positive selection. Also the drift term I refer to in the case of engraftment is due to sampling bias of what one transplants, it is not the same drift occurring in human tumours that it is due to random cell death in small populations.

Reviewer #3:

Reviewer#3 suggests to the editor acceptance of the manuscript in this revised form despite the lack of WGS and computational systems biology approach.

We appreciate very much the extremely valuable comments of the reviewer 2 and his/her suggestions. Specifically, we addressed the major concern regarding the variant calling and re-analyzed the data using more stringent criteria (see methods section). As a consequence of the new stringent criteria, the absolute numbers of detected mutations and neoantigens are now different from the previous version. However, the major findings of our study remain the same.

The changes that were made in the manuscript since the first revised version are marked in red and the point-by-point responses to the reviewer are in the following pages.

We hope that these revisions will satisfy the criticisms of the reviewer.

Reviewer #2 (Remarks to the Author):

Although the authors did attempt to address my comments, there are quite a few important points of concern that need to be addressed, especially regarding the new data presented on the CT26 cell line:

1) page 9

here the authors present a new analysis using the same approach but on a different cell line (CT26). What I find completely puzzling, is that the new cell line has the same subclonal composition than MC38, with a large subclonal cluster at around 70% CCF. This cannot possibly be right, if I was dubious of the author's explanation of a complex subclonal architecture and branching structure in MC38, now I am convinced there must be something wrong in the variant calling as such complex structure cannot be the same also in CT26, particularly because the individual variants are completely different. Maybe there is something wrong in the reference sample used. The authors must address this problem!

We appreciate very much the important issues pointed out by the reviewer. In order to obtain high confidence mutations, we reanalyzed the samples using more stringent filtering. We performed trimming of the reads so that the lower quality bases at the beginning and end of the reads are removed. Then we aligned the reads using BWA instead of Tmap. Finally, we filtered out all mutations with less than 10 alternative reads. The new VAF plots (Figure S10) show high number of mutations with VAF around 0.5. Additionally, since we cannot obtain allele-specific CNAs, instead of estimating the mutation copy number, we used a different method to classify the mutations into clonal and subclonal (page 16). All figures resulting from these changes were modified accordingly.

2) row 311-313

the ratio of expressed neoantigens and total number of exonic mutations should DECREASE under immunoediting because the immune system gets rid of mutations generating neoantigens, but not the others...

According to our analyses (Angelova et al., Genome Biology 2015, Charoentong et al., Cell Reports 2017, Mlecnik et al., Immunity 2017) and analyses by others using algorithms to predict neoantigens (e.g. Rooney et al., Cell 2015), the majority of the mutations are generating neoantigens. Hence, a strong immunoediting would not necessarily imply that the neoantigen/mutation ratio would decrease. However, as of today, it is not possible to determine which neoantigens are immunogenic and are targeted by the immune system. While direct identification of neoantigens using mass spectrometry might provide possible answers, the sensitivity of the technique is currently too low to provide a definitive answer.

3) row 316

why in the anti-PDL1 there are MORE clonal expressed neoantigens? Immunoediting would predict there should be less

The results of the more stringent analyses in the second revision showed that there are slightly more clonal neoantigens in the anti-PDL1 samples (95% vs. 93%). In line with the above argumentation, it is currently not known if the immune system is preferentially targeting clonal or subclonal neoantigens. While some reports indicate that this might be the case following therapy with checkpoint blockers (McGranahan et al, Science 2016), further studies with larger number of samples are necessary.

4) row 321

if the hypothesis of immunoediting is correct, then the CCF shift should be due to negative selection for a lot of immunogenic subclones and consequently the rise in frequency of the non-immunogenic ones, hence this is rather negative selection, not positive (Darwinian) selection.

We completely agree with the reviewer that there is a negative selection of the immunogenic subclones, and modified the sentence appropriately.

5) *in the abstract, rows 47-54*

here do the authors just mean that when tumours are hiding from the immune system (e.g. through upregulation of PD1), then immunoediting is weak and the neutral accumulation of mutations dominates? After this initial sentence, do they mean immunoediting start playing a role when PD1 is suppressed? If this is the case, they should spell it out more clearly in the abstract as I struggled to get it. I would also avoid the general statement that “neutral evolution is THE major force sculpting the tumour” as even in the case of a neutrally evolving tumour, the past there must have selection for driver alterations.

We thank the reviewer for this comment and modified the abstract to be more clearer.

6) *rows 195-199*

the authors seem to suggest that the upregulation of PD1 occurs without a selective sweep for high-PD1 clones, but rather through plasticity of cells switching to a high-PD1 program. Here I would avoid explicitly refer to bet-hedging as it would not resonate with the general readership and just state that the upregulation of PD1 is likely due to cell phenotypic plasticity, not positive selection for high-PD1 clones.

We thank the reviewer for the very useful suggestion. We have taken it into account and modified the specific sentence as suggested.

7) *figure 3A*

the venn diagrams are not weighted and do not convey the message that most mutations are in common to all three cases

We agree with the reviewer on this point and we made an attempt to visualize the data accordingly. However, because of the high percentage of shared vs non-shared mutations, the weighted venn diagrams are heavily skewed and make it difficult to provide the complete information. We therefore decided to use the simpler figure.

8) *row 224*

can please the authors refer to a venn diagram to illustrate the mutations shared between the replicas?

The reference to the sample specific Venn diagrams have been added on page 6, second paragraph.

9) *row 227-230*

I don't understand this sentence at all

We appreciate the reviewer's comment and simplified the paragraph by deleting the sentence.

10) *row 283-284*

not sure what the authors mean here, it seems they simply do not observe positive (Darwinian) selection because the subclonal architecture does not change over time

We appreciate also very much this comment and have modified the sentence and now it states that the clonal dynamics of this cancer cell line over time is not dominated by strong Darwinian selection, but rather follows neutral evolution.

11) *row 420*

in the discussion the authors reported my comments on the driving evolutionary forces at play here, they should note that this refers to their experimental setting, not generally to evolution of human tumours, where there is no engraftment of course, and there could be positive selection. Also the drift term I refer to in the case of engraftment is due to sampling bias of what one transplants, it is not the same drift occurring in human tumours that it is due to random cell death in small populations.

We thank the reviewer for the comment and we have modified the discussion accordingly.

Reviewer #2 (Remarks to the Author):

The authors addressed all my comments.